# StableSSM: Alleviating the Curse of Memory in State-space Models through Stable reparameterization

## Abstract

In this paper, we investigate the long-term memory learning capabilities of state-space models (SSMs) from the perspective of parameterization. We prove that state-space models without any reparameterization exhibit a memory limitation similar to that of traditional RNNs: the target relationships that can be stably approximated by state-space models must have an exponential decaying memory. Our analysis identifies this "curse of memory" as a result of the recurrent weights converging to a stability boundary, suggesting that a reparameterization technique can be effective. To this end, we introduce a class of reparameterization techniques for SSMs that effectively lift its memory limitations. Besides improving approximation capabilities, we further illustrate that a principled choice of reparameterization scheme can also enhance optimization stability. We validate our findings using synthetic datasets and language models.

## 1 Introduction

Understanding long-term memory relationships is fundamental in sequence modeling. Capturing this prolonged memory is vital, especially in applications like time series prediction (Connor et al., 1994), language models (Sutskever et al., 2011; Fu et al., 2023; Poli et al., 2023). Since its emergence, transformers (Vaswani et al., 2017) have become the go-to models for language representation tasks (Brown et al., 2020). However, a significant drawback lies in their computational complexity, which is asymptotically $O(T^2)$, where $T$ is the sequence length. This computational bottleneck has been a critical impediment to the further scaling-up of transformer models. State-space models such as S4 (Gu et al., 2021), S5 (Smith et al., 2023), RWKV (Peng et al., 2023) and RetNet (Sun et al., 2023) offer an alternative approach. These models are of the recurrent type and excel in long-term memory learning. Their architecture is specifically designed to capture temporal dependencies over extended sequences, providing a robust solution for tasks requiring long-term memory (Tay et al., 2021). One of the advantages of state-space models over traditional RNNs lies in their computational efficiency, achieved through the application of parallel scan algorithms (Martin & Cundy, 2018). Traditional nonlinear RNNs are often plagued by slow forward and backward propagation, a limitation that state-space models circumvent by leveraging linear RNN blocks.

As traditional nonlinear RNNs exhibit an asymptotically exponential decay in memory (Wang et al., 2023), the SSMs variants like S4 overcome some of the memory issues. The previous empirical results suggest that either (i) the "linear dynamics and nonlinear layerwise activation" or (ii) the parameterization inherent to S4, is pivotal in achieving the enhanced performance. Current research answers which one is more important. We first prove an inverse approximation theorem showing that state-space models without reparameterization still suffer from the "curse of memory", which is consistent with empirical results (Wang & Xue, 2023). This rules out the point (i) as the reason for SSMs' good long-term memory learning. A natural question arises regarding whether the reparameterizations are the key to learn long-term memory. We prove a class of reparameterization functions $f$, which we call stable reparameterization, enables the stable approximation of linear functionals. This includes commonly used exponential reparameterization and softplus reparameterization. Furthermore, we question whether S4's parameterizations are optimal. Here we give a particular sense in terms of optimization that they are not optimal. We propose the optimal one and show its stability via numerical experiments.

We summarize our main contributions as follow:

1. We prove that similar to RNNs, the state-space models without reparameterization can only stably approximate targets with exponential decaying memory.
2. We identify a class of stable reparameterization which achieves the stable approximation of **any** linear functionals. Both theoretical and empirical evidence highlight that stable reparameterization is crucial for long-term memory learning.
3. From the optimization viewpoint, we propose a criterion based on the gradient-over-weight ratio. We identify the "best" reparameterization in this sense and verify its performance.

**Notation.** We use the bold face to represent the sequence while then normal letters are scalars, vectors or functions. Throughout this paper we use $\| \cdot \|$ to denote norms over sequences of vectors, or function(al)s, while $| \cdot |$ (with subscripts) represents the norm of number, vector or weights tuple. Here $|x|_\infty := \max_i |x_i|, |x|_2 := \sqrt{\sum_i x_i^2}, |x|_1 := \sum_i |x_i|$ are the usual max ($L_\infty$) norm, $L_2$ norm and $L_1$ norm. We use $m$ to denote the hidden dimension.

## 2 BACKGROUND

In this section, we first introduce the state-space models and compare them to traditional nonlinear RNNs. Subsequently, we adopt the sequence modeling as a problem in nonlinear functional approximation framework. Specifically, the theoretical properties we anticipate from the targets are defined. Moreover, we define the "curse of memory" phenomenon and provide a concise summary of prior theoretical definitions and results concerning RNNs.

### 2.1 STATE-SPACE MODELS

Recurrent neural networks (RNNs) (Rumelhart et al., 1986) are a family of neural networks specialized in sequence modeling. With trainable weights $W \in \mathbb{R}^{m \times m}, U \in \mathbb{R}^{m \times d}, b, c \in \mathbb{R}^m$ and activation function $\sigma(\cdot)$, the simplest RNN maps $d$-dimensional input sequence $\mathbf{x} = \{x_t\}$ to 1-dimensional output sequence $\{\hat{y}_t\}$. To simplify our analysis, we utilize the continuous-time framework referenced in Li et al. (2020):

$$
\begin{aligned}
\frac{dh_t}{dt} &= \boldsymbol{\sigma}(W h_t + U x_t + b) & h_{-\infty} &= 0, \\
\hat{y}_t &= c^\top h_t, & t &\in \mathbb{R}.
\end{aligned}
\tag{1}
$$

State-space models refer to the type of neural networks with layer-wise nonlinearity but linear dynamics in the hidden states. As detailed in Appendix E, the following form is a simplification of practical SSMs in the sense that practical SSMs can be realized by the stack of Equation (2).

$$
\begin{aligned}
\frac{dh_t}{dt} &= W h_t + U x_t + b, & h_{-\infty} &= 0, \\
\hat{y}_t &= c^\top \boldsymbol{\sigma}(h_t), & t &\in \mathbb{R}.
\end{aligned}
\tag{2}
$$

It is known that multi-layer state-space models are universal approximators (Orvieto et al., 2023; Wang & Xue, 2023). In particular, when the nonlinearity is added layer-wise, it is sufficient (in approximation sense) to use *real diagonal W* (Gu et al., 2022; Li et al., 2022). In this paper, we only consider the real diagonal matrix case and denote it by $\Lambda$.

$$
\frac{dh_t}{dt} = \Lambda h_t + U x_t + b
\tag{3}
$$

Compared with S4, the major differences lies in initialization such as HiPPO (Gu et al., 2020) and parameters saving method such as DPLR (Gu et al., 2022) and NPLR (Gu et al., 2021).

We assume the hidden states remain uniformly bounded for any input sequence $\mathbf{x}$, irrespective of the hidden dimensions $m$. Specifically, this can be expressed as

$$
\sup_m \sup_t |h_t|_\infty < \infty.
\tag{4}
$$

Also, we assume the weights are uniformly bounded with $\sup_m \max(|c|_2, |\Lambda|_2, |U|_2, |b|_2) < \infty$. We focus on strictly increasing, continuously differentiable nonlinear activations with Lipschitz constant $L_0$. This property holds for activations such as tanh, sigmoid, softsign $\sigma(z) = \frac{z}{1+|z|}$.

## 2.2 Sequence Modeling as Nonlinear Functional Approximations

Sequential modeling aims to discern the association between an input series, represented as $\mathbf{x} = \{x_t\}$, and its corresponding output series, denoted as $\mathbf{y} = \{y_t\}$. The input series are continuous bounded inputs vanishing at infinity: $\mathbf{x} \in \mathcal{X} = C_0(\mathbb{R}, \mathbb{R}^d)$ with norm $\|\mathbf{x}\|_\infty := \sup_{t \in \mathbb{R}} |x_t|_\infty$. It is assumed that the input and output sequences are determined from the inputs via a set of functionals, symbolized as $\mathbf{H} = \{H_t : \mathcal{X} \to \mathbb{R} : t \in \mathbb{R}\}$, through the relationship $y_t = H_t(\mathbf{x})$. In essence, the challenge of sequential approximation boils down to estimating the desired functional sequence $\mathbf{H}$ using a different functional sequence $\widehat{\mathbf{H}}$ potentially from a predefined model space such as SSMs.

We first introduce the definitions on (sequences of) functionals as discussed in (Wang et al., 2023).

**Definition 2.1.** Let $\mathbf{H} = \{H_t : \mathcal{X} \mapsto \mathbb{R}; t \in \mathbb{R}\}$ be a sequence of functionals.

1. (**Linear**) $H_t$ is linear functional if for any $\lambda, \lambda' \in \mathbb{R}$ and $\mathbf{x}, \mathbf{x}' \in \mathcal{X}$, $H_t(\lambda \mathbf{x} + \lambda' \mathbf{x}') = \lambda H_t(\mathbf{x}) + \lambda' H_t(\mathbf{x}')$.

2. (**Continuous**) $H_t$ is continuous functional if for any $\mathbf{x},' \mathbf{x} \in \mathcal{X}$, $\lim_{\mathbf{x}' \to \mathbf{x}} |H_t(\mathbf{x}') - H_t(\mathbf{x})| = 0$.

3. (**Bounded**) $H_t$ is bounded functional if the norm of functional $\|H_t\|_\infty := \sup_{\{\mathbf{x} \neq 0\}} \frac{|H_t(\mathbf{x})|}{\|\mathbf{x}\|_\infty + 1} + |H_t(\mathbf{0})| < \infty$.

4. (**Time-homogeneous**) $\mathbf{H} = \{H_t : t \in \mathbb{R}\}$ is time-homogeneous (or shift-equivariant) if the input-output relationship commutes with time shift: let $[S_\tau(\mathbf{x})]_t = x_{t-\tau}$ be a shift operator, then $\mathbf{H}(S_\tau \mathbf{x}) = S_\tau \mathbf{H}(\mathbf{x})$.

5. (**Causal**) $H_t$ is causal functional if it does not depend on future values of the input. That is, if $\mathbf{x}, \mathbf{x}'$ satisfy $x_t = x_t'$ for any $t \leq t_0$, then $H_t(\mathbf{x}) = H_t(\mathbf{x}')$ for any $t \leq t_0$.

6. (**Regular**) $H_t$ is regular functional if for any sequence $\{\mathbf{x}^{(n)} : n \in \mathbb{N}\}$ such that $x_s^{(n)} \to 0$ for almost every $s \in \mathbb{R}$, then $\lim_{n \to \infty} H_t(\mathbf{x}^{(n)}) = 0$.

The continuity, boundedness, time-homogeneity, causality are important properties for good sequence-to-sequence models to have. Linearity is an important simplification as many theoretical theorems are available in functional analysis. Without loss of generality, we assume that the nonlinear functionals satisfy $H_t(\mathbf{0}) = 0$. It can be achieved via studying $H_t^{\text{adjusted}}(\mathbf{x}) = H_t(\mathbf{x}) - H_t(\mathbf{0})$.

## 2.3 Curse of Memory and Stable Approximation

The concept of memory has been extensively explored in academic literature, yet much of this work relies on heuristic approaches and empirical testing, particularly in the context of learning long-term memory (Poli et al., 2023). Here we study the memory property from a theoretical perspective.

The "curse of memory" phenomenon, which was originally formulated for linear functionals and linear RNNs, is well-documented in prior research (Li et al., 2020; 2022). Building upon this foundation, our study employs the extended framework proposed by Wang et al. (2023), which specifically focuses on nonlinear RNNs. However, these studies do not address the case of state-space models. Within the same framework, the slightly different memory function and decaying memory concepts enable us to explore the approximation capabilities of nonlinear functionals using SSMs. We add 1 in the memory function definition to make it more regular.

**Definition 2.2** (Memory function and decaying memory). For bounded, causal, continuous, regular and time-homogeneous nonlinear functional sequences $\mathbf{H} = \{H_t : t \in \mathbb{R}\}$ on $\mathcal{X}$, define the following function as the *memory function* of $\mathbf{H}$: Over bounded Heaviside input $\mathbf{u}^x(t) = x \cdot \mathbf{1}_{\{t \geq 0\}}$

$$\mathcal{M}(\mathbf{H})(t) := \sup_{x \neq 0} \frac{\left| \frac{d}{dt} H_t(\mathbf{u}^x) \right|}{|x|_\infty + 1}. \tag{5}$$

We assume the memory function of the target is finite for all $t \in \mathbb{R}$. The functional sequences $\mathbf{H}$ have a *decaying memory* if $\lim_{t \to \infty} \mathcal{M}(\mathbf{H})(t) = 0$. In particular, we say it has an *exponential (polynomial) decaying memory* if there exists constant $\beta > 0$ such that $\lim_{t \to \infty} e^{\beta t} \mathcal{M}(\mathbf{H})(t) = 0$ ($\lim_{t \to \infty} t^\beta \mathcal{M}(\mathbf{H})(t) = 0$).

Similar to Wang et al. (2023), this adjusted memory function definition is also compatible with the memory concept in linear functional which is based on the famous Riesz representation theorem (Theorem A.2). As shown in Appendix B.1, the nonlinear functionals constructed by state-space models are point-wise continuous over Heaviside inputs. Combined with time-homogeneity, we know that state-space models are nonlinear functionals with decaying memory (see Appendix B.2).

**Definition 2.3** (Functional sequence approximation in Sobolev-type norm). Given functional sequences $\mathbf{H}$ and $\widehat{\mathbf{H}}$, we consider the approximation in the following Sobolev-type norm:

$$\left\| \mathbf{H} - \widehat{\mathbf{H}} \right\|_{W^{1,\infty}} = \sup_t \left( \| H_t - \widehat{H}_t \|_\infty + \left\| \frac{dH_t}{dt} - \frac{d\widehat{H}_t}{dt} \right\|_\infty \right). \tag{6}$$

**Definition 2.4** (Perturbation error). For target $\mathbf{H}$ and parameterized model $\widehat{\mathbf{H}}(\cdot, \theta_m), \theta_m \in \Theta_m := \{\mathbb{R}^{m \times m} \times \mathbb{R}^{m \times d} \times \mathbb{R}^m \times \mathbb{R}^m\}$, we define the *perturbation error* for hidden dimension $m$:

$$E_m(\beta) := \sup_{\tilde{\theta}_m \in \{\theta : |\theta - \theta_m|_2 \leq \beta\}} \| \mathbf{H} - \widehat{\mathbf{H}}(\cdot; \tilde{\theta}_m) \|. \tag{7}$$

In particular, $\widetilde{\mathbf{H}}$ refers to the perturbed models $\widehat{\mathbf{H}}(\cdot; \tilde{\theta}_m)$. Moreover, $E(\beta) := \limsup_{m \to \infty} E_m(\beta)$ is the asymptotic perturbation error. The weight norm for SSM is $|\theta|_2 := \max(|\Lambda|_2, |U|_2, |b|_2, |c|_2)$.

Based on the definition of perturbation error, we consider the stable approximation as introduced by Wang et al. (2023).

**Definition 2.5** (Stable approximation). Let $\beta_0 > 0$. We say a target functional sequence $\mathbf{H}$ admits a $\beta_0$-*stable approximation* if for all $0 \leq \beta \leq \beta_0$, the perturbed error satisfies that: (i) $E(0) = 0$; (ii) $E(\beta)$ is continuous for $\beta \in [0, \beta_0]$.

Equation $E(0) = 0$ means the functional sequence is an approximation. As the stable approximation is the necessary requirement for the optimal parameters to be found by the gradient-based optimizations, it is a desirable assumption.

The "curse of memory" describes the phenomenon where targets approximated by linear, hardtanh, or tanh RNNs must demonstrate an exponential decaying memory. However, empirical observations suggest that state-space Models, particularly the S4 variant, may possess favorable properties. Thus, it is crucial to ascertain whether the inherent limitations of RNNs can be circumvented using state-space models. Given the impressive performance of state-space models, notably S4, a few pivotal questions arise: Do the model structure of state-space models overcome the "curse of memory"? In the subsequent section, we will demonstrate that the model structure of state-space models does not indeed address the curse of memory phenomenon.

## 3 Main results

In this section, we first prove that similar to the traditional recurrent neural networks (Li et al., 2020; Wang et al., 2023), state-space models without reparameterization suffer from the "curse of memory" problem. This implies the targets that can be stably approximated by SSMs must have exponential decaying memory. Our analysis reveals that the problem arises from recurrent weights converging to a stability boundary when learning targets associated with long-term memory. Therefore, we introduce a class of stable reparameterization techniques to achieve the stable approximation for targets with polynomial decaying memory.

Beside the benefit of approximation perspective, we also discuss the optimization benefit of the stable reparameterizations. We show that the stable reparameterization can make the gradient scale more balanced, therefore the optimization of large models can be more stable.

### 3.1 SSMs are not stable approximators

In this section, we present a theoretical theorem demonstrating that the state-space structure does not alleviate the "curse of memory" phenomenon. State-space models consist of alternately stacked linear RNNs and nonlinear activations. Our proof is established for the shallow case. As recurrent models, SSMs without reparameterization continue to exhibit the commonly observed phenomenon of exponential memory decay, as evidenced by empirical findings (Wang & Xue, 2023).

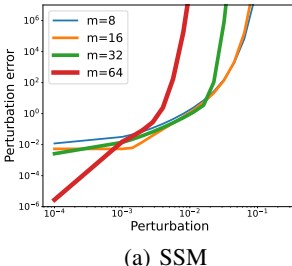
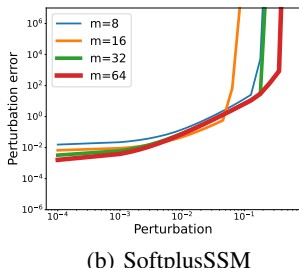
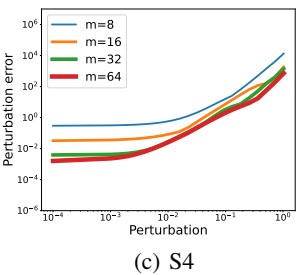

|                |                    |           |
| :------------: | :----------------: | :-------: |
| (a) SSM        | (b) SoftplusSSM    | (c) S4    |

Figure 1: State-space models without stable reparameterization cannot approximate targets with polynomial decaying memory. In (a), the intersection of lines are shifting towards left as the hidden dimension $m$ increases. In (b), SSMs using softplus reparameterization has a stable approximation. In (c), S4 can stably approximate the target with better stability.

**Theorem 3.1** (Curse of memory in SSMs). *Assume $\mathbf{H}$ is a sequence of bounded, causal, continuous, regular and time-homogeneous functionals on $\mathcal{X}$ with decaying memory. Suppose there exists a sequence of state-space models $\{\widehat{\mathbf{H}}(\cdot, \theta_m)\}_{m=1}^{\infty}$ $\beta_0$-stably approximating $\mathbf{H}$ in the norm defined in Equation (6). Assume the model weights are uniformly bounded: $\theta_{\max} := \sup_m |\theta_m|_2 < \infty$. Then the memory function $\mathcal{M}(\mathbf{H})(t)$ of the target decays exponentially:*

$$\mathcal{M}(\mathbf{H})(t) \leq (d+1)L_0\theta_{\max}^2 e^{-\beta t}, \quad t \geq 0, \beta < \beta_0. \tag{8}$$

The proof of Theorem 3.1 is provided in Appendix B.3. The stability boundary is discussed in Remark B.1. Compared with previous results (Li et al., 2020; Wang et al., 2023), the main proof difference comes from Lemma B.6 as the activation is in the readout $y_t = c^{\top}\sigma(h_t)$. Our results provide a more accurate characterization of memory decay, in contrast to previous works that only offer qualitative estimates. A consequence of Theorem 3.1 is that if the target exhibits a non-exponential decay (e.g., polynomial decay), the recurrent weights converge to a stability boundary, thereby making the approximation unstable. Finding optimal weights can become challenging with gradient-based optimization methods, as the optimization process tends to become unstable with the increase of model size. The numerical verification is presented in Figure 1 (a). The lines intersect and the intersections points shift towards the 0, suggesting that the stable radius $\beta_0$ does not exist. Therefore SSMs without reparameterization cannot stably approximate targets with polynomial decaying memory.

### 3.2 STABLE REPARAMETERIZATIONS AND THE APPROXIMATION BENEFIT

The proof of Theorem 3.1 suggests that the "curse of memory" arises due to the recurrent weights approaching a stability boundary. Additionally, our numerical experiments (in Figure 1 (c)) show that while state-space models suffer from curse of memory, the commonly used S4 layer (with exponential) ameliorates this issue. However, it is not a unique solution. Our findings highlight that the foundation to achieving a stable approximation is the stable reparameterization method, which we define as follows:

**Definition 3.2** (Stable reparameterization). We say a reparameterization scheme $f : \mathbb{R} \to \mathbb{R}$ is stable if there exists a continuous function $g$ such that: $g : [0, \infty) \to [0, \infty), g(0) = 0$:

$$\sup_w \left[ |f(w)| \sup_{|\tilde{w}-w| \leq \beta} \int_0^\infty \left| e^{f(\tilde{w})t} - e^{f(w)t} \right| dt \right] \leq g(\beta). \tag{9}$$

For example, commonly used reparameterization (Gu et al., 2021; Smith et al., 2023) such as $f(w) = -e^w$, $f(w) = -\log(1 + e^w)$ are all stable. Verifications are provided in Remark B.3.

As depicted in Figure 1 (b), state-space models with stable reparameterization can approximate targets exhibiting polynomial decay in memory. In particular, we prove that under a simplified perturbation setting (solely perturbing the recurrent weights), any linear functional can be stably approximated by linear RNNs. This finding under simplified setting is already significant as the instability in learning long-term memory mainly comes from the recurrent weights.

**Theorem 3.3.** *For **any** bounded, causal, continuous, regular, time-homogeneous linear functional* **H***, assume* **H** *is approximated by a sequence of linear RNNs* $\{\widehat{\mathbf{H}}(\cdot, \theta_m)\}_{m=1}^{\infty}$ *with stable reparameterization, then this approximation is a stable approximation.*

The proof of Theorem 3.3 can be found in Appendix B.4. Compared to Theorem 3.1, Theorem 3.3 underscores the pivotal role of stable reparameterization in achieving stable approximation of linear functional with long-term memory.

### 3.3 OPTIMIZATION BENEFIT OF STABLE REPARAMETERIZATION

In the previous section, the approximation benefit of stable reparameterization in SSMs is discussed. Therefore, a natural question is whether the commonly used reparameterizations in S4/S5 are the optimal ones. Here we give a optimization criterion and show that they are not optimal. As pointed out by Li et al. (2020; 2022), the approximation of linear functionals using linear RNNs can be reduced into the approximation of $L_1$-integrable memory function $\rho(t)$ via functions of the form $\hat{\rho}(t) = \sum_{i=1}^{m} c_i e^{-\lambda_i t}$.

$$\rho(t) \approx \sum_{i=1}^{m} c_i e^{-\lambda_i t}, \quad \lambda_i > 0. \tag{10}$$

Within this framework, $\lambda_i$ is interpreted as the decay mode. Approaching this from the gradient-based optimization standpoint, and given that learning rates are shared across different decay modes, a fitting characterization for "good parameterization" emerges: *The gradient scale across different memory decays modes should be Lipschitz continuous with respect to the weights scale.*

$$|\text{Gradient}| := \left| \frac{\partial \text{Loss}}{\partial \lambda_i} \right| \le L|\lambda_i|. \tag{11}$$

The Lipschitz constant is denoted by $L$. Without this property, the optimization process can be sensitive to the learning rate. We give a detailed discussion in Appendix F. In the following theorem, we characterize the relationship between gradient norms and recurrent weight parameterization.

**Theorem 3.4** (Parameterizations influence the gradient norm scale). *Assume the target functional sequence* **H** *is being approximated by a sequence of SSMs* $\widehat{\mathbf{H}}_m$. *Assume the (diagonal) recurrent weight matrix is parameterized via* $f : \mathbb{R} \to \mathbb{R} : f(w) = \lambda$. $w$ *is the trainable weight while* $\lambda$ *is the eigenvalue of recurrent weight matrix* $\Lambda$. *The gradient norm* $G_f(w)$ *is upper bounded by the following expression:*

$$G_f(w) := \left| \frac{\partial Loss}{\partial w} \right| \le C_{\mathbf{H}, \widehat{\mathbf{H}}_m} \frac{|f'(w)|}{f(w)^2}. \tag{12}$$

*Here* $C_{\mathbf{H}, \widehat{\mathbf{H}}_m}$ *is independent of the parameterization* $f$ *provided that* **H**, $\widehat{\mathbf{H}}_m$ *are fixed. Moreover, the discrete-time version is* $G_f^D(w) := \left| \frac{\partial Loss}{\partial w} \right| \le C_{\mathbf{H}, \widehat{\mathbf{H}}_m} \frac{|f'(w)|}{(1-f(w))^2}$.

Refer to Appendix B.5 for the proof of Theorem 3.4. Here we discuss the feasibility to extend to deeper networks. As the reparameterization only change the gradients of state-space models, so the gradient analysis holds for the recurrent weights in complex models which will also be bounded by a parameterization-dependent term in Equation (12). Therefore Theorem 3.4 will also be effective for multi-layer networks. In Table 1 we summarize common reparameterization methods and corresponding gradient scale functions. Moreover, according to the criterion given in Equation (11), the "best" stable reparameterization should satisfy $G_f(w) \equiv L|w|$ for some constant $L > 0$. The equation can be solved into the following form: $f(w) = \frac{1}{aw^2+b}, a \ne 0, b \in \mathbb{R}$ (proof is given in Appendix G). Similarly, the discrete case gives the solution $f(w) = 1 - \frac{1}{aw^2+b}$. We choose $a = 1, b = 0.5$ because this ensures the stability of the hidden state dynamics and stable approximation in Equation (9). The stability of linear RNN further requires $a > 0$ and $b \ge 0$. We plot and compared $\frac{G_f(w)}{|w|}$ of different stable reparameterizations in Figure 2. It can be seen that, compared with linear and exponential reparameterizations, the softplus reparameterization is generally milder in this gradient-over-weight criterion. The "best" parameterization is optimal in the sense it has a balanced gradient-over-weight ratio across different weights.

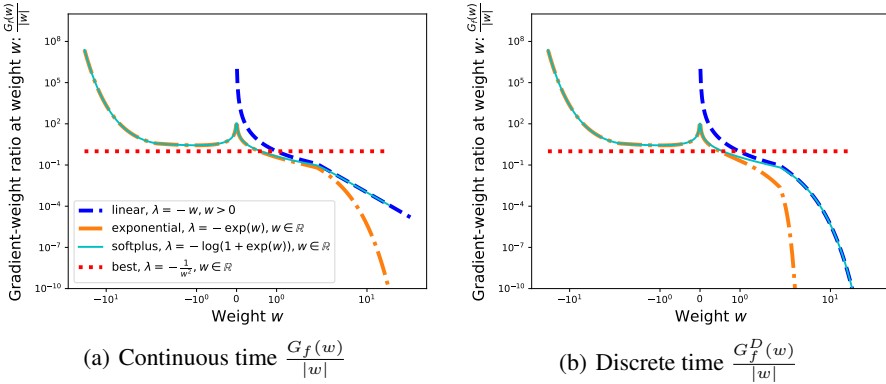

(a) Continuous time $\frac{G_f(w)}{|w|}$       (b) Discrete time $\frac{G_f^D(w)}{|w|}$

Figure 2: Gradient norm function $G_f$ and $G_f^D$ of different parameterization methods. The "best" parameterization methods maintain a balanced gradient-over-weight ratio.

Table 1: Summary of reparameterizations and corresponding gradient norm functions in continuous and discrete time. Notice that the $G_f$ and $G_f^D$ are rescaled up to a constant $C_{\mathbf{H},\widehat{\mathbf{H}}}$.

|  | Reparameteriations | $f$ | $G_f$ or $G_f^D$ |
|---|---|---|---|
| Continuous | ReLU | $-\text{ReLU}(w)$ | $\frac{1}{w^2}\mathbf{1}_{\{w>0\}}$ |
|  | Exp | $-\exp(w)$ | $e^{-w}$ |
|  | Softplus | $-\log(1+\exp(w))$ | $\frac{\exp(w)}{(1+\exp(w))\log(1+\exp(w))^2}$ |
|  | "Best"(Ours) | $-\frac{1}{aw^2+b}, a>0, b>0$ | $2a|w|$ |
| Discrete | ReLU | $\exp(-\text{ReLU}(w))$ | $\frac{\exp(-w)}{(1-\exp(-w))^2}\mathbf{1}_{\{w>0\}}$ |
|  | Exp | $\exp(-\exp(w))$ | $\frac{\exp(w-\exp(w))}{(1-\exp(-\exp(w)))^2}$ |
|  | Softplus | $\frac{1}{1+\exp(w)}$ | $e^{-w}$ |
|  | Tanh | $\tanh(w)=\frac{e^{2w}-1}{e^{2w}+1}$ | $e^{2w}$ |
|  | "Best"(Ours) | $1-\frac{1}{w^2+0.5}\in(-1,1)$ | $2|w|$ |

## 4 NUMERICAL VERIFICATIONS

Based on the above analyses, we verify the theoretical statements over synthetic tasks and language models using WikiText-103. The additional numerical details are provided in Appendix C.

### 4.1 SYNTHETIC TASKS

Linear functionals have a clear structure, allowing us to study the differences of parameterizations. Similar to Li et al. (2020) and Wang et al. (2023), we consider linear functional targets $\mathbf{H}$ with following polynomial memory function $\rho(t)=\frac{1}{(t+1)^{1.1}}$: $y_t=H_t(\mathbf{x})=\int_{-\infty}^{t}\rho(t-s)x_s ds$. We use the state-space models with tanh activations to learn the sequence relationships. In Figure 3, the eigenvalues $\lambda$ are initialized to be the same while the only difference is the reparameterization function $f(w)$. Training loss across different reparameterization schemes are similar but the gradient-over-weight ratio across different parameterization schemes are different in terms of the scale.

### 4.2 LANGUAGE MODELS

In addition to the synthetic dataset of linear functions, we further justify Theorem 3.4 by examining the gradient-over-weight ratios for language models using state-space models. In particular, we adopt the Hyena (Poli et al., 2023) architecture while the implicit convolution is replaced by a simple real-weighted state-space model (Smith et al., 2023).

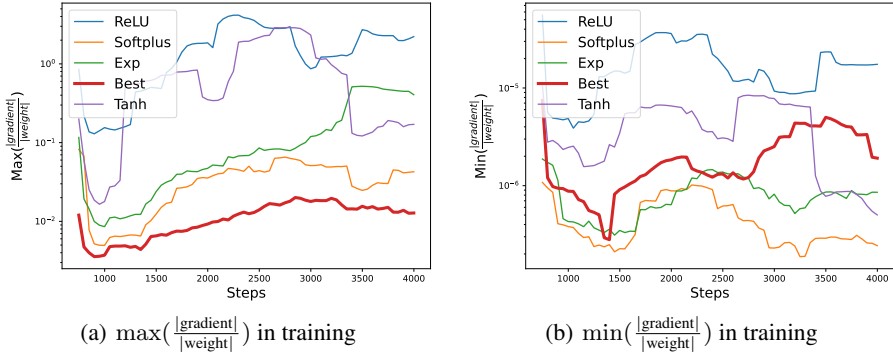

(a) $\max(\frac{|\text{gradient}|}{|\text{weight}|})$ in training

(b) $\min(\frac{|\text{gradient}|}{|\text{weight}|})$ in training

Figure 3: In the learning of linear functionals of polynomial decaying memory, the gradient-over-weight scale range during the training of state-space models. It can be seen the "best"discrete parameterization $f(w) = 1 - \frac{1}{w^2+0.5}$ achieves the smallest gradient-over-weight scale. 0.5 is added to ensure $f(w) \in (-1, 1)$. Such property is desirable when a large learning rate is used in training.

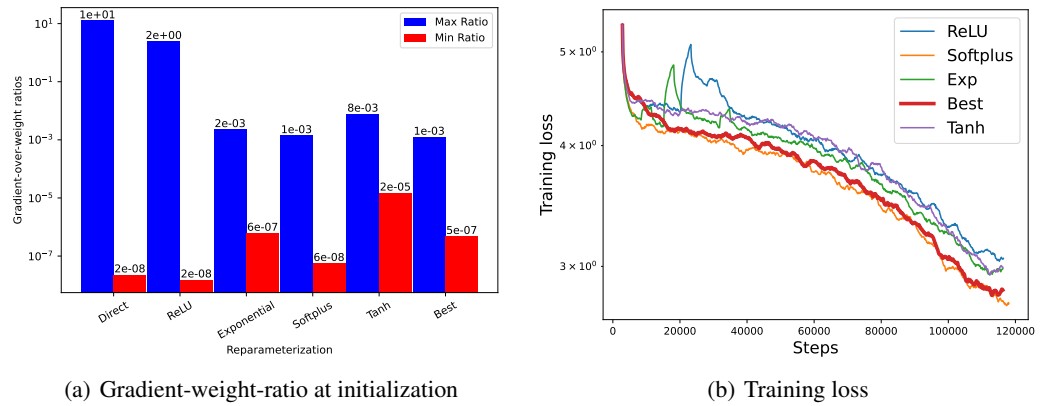

(a) Gradient-weight-ratio at initialization

(b) Training loss

Figure 4: Language models on WikiText-103. In (a), we show the gradient-over-weight ratio ranges for different parameterizations of recurrent weights in state-space models. The eigenvalues $\lambda$ are initialized to be the same while the only difference is the reparameterization function $f$. In (b), the "Best" parameterization is more stable than the ReLU and exponential reparameterizations. Additional experiments for different learning rates are provided in Figure 6.

In Figure 4 (a), given the same initialization, we show that stable reparameterizations such as exponential, softplus, tanh and "best" exhibit a narrower range of gradient-over-weight ratios compared to both the direct and relu reparameterizations. Beyond the gradient at the same initialization, in Figure 5, we show the gradient-over-weight ratios during the training process. The stable reparameterization will give better gradient-over-weight ratios in the sense that the "best" stable reparameterization maintains the smallest $\max(\frac{|\text{grad}|}{|\text{weight}|})$. Specifically, as illustrated in Figure 4 (b) and Figure 6, while training with a large learning rate may render the exponential parameterization unstable, the "best" reparameterization $f(w) = 1 - \frac{1}{w^2+0.5}$ appears to enhance training stability.

## 5 RELATED WORK

RNNs, as introduced by Rumelhart et al. (1986), represent one of the earliest neural network architectures for modeling sequential relationships. Empirical findings by Bengio et al. (1994) have shed light on the challenge of exponential decaying memory in RNNs. Various works (Hochreiter & Schmidhuber, 1997; Rusch & Mishra, 2022; Wang & Yan, 2023) have been done to improve the memory patterns of recurrent models. Theoretical approaches (Li et al., 2020; 2022; Wang et al.,

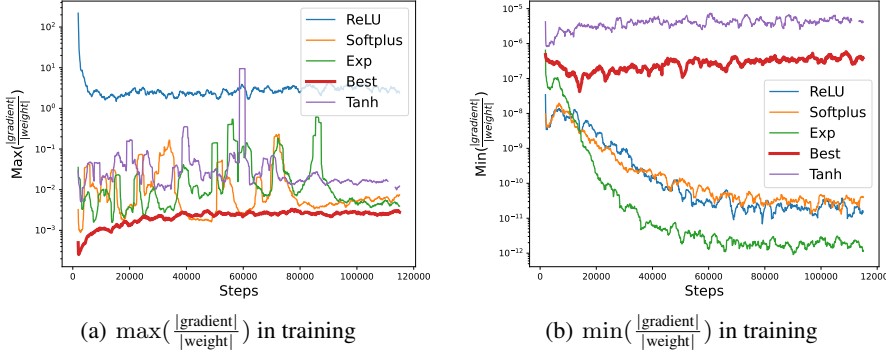

(a) $\max(\frac{|\text{gradient}|}{|\text{weight}|})$ in training

(b) $\min(\frac{|\text{gradient}|}{|\text{weight}|})$ in training

Figure 5: Gradient-over-weight ratio for different reparameterizations of the recurrent weights in language modeling. The "best" reparameterization $f(w) = 1 - \frac{1}{w^2 + 0.5}$ maintains the smallest $\max(\frac{|\text{grad}|}{|\text{weight}|})$ which is crucial for the training stability as illustrated in Figure 4 (b).

2023) have been taken to study the exponential memory decay of RNNs. In this paper, we study the state-space models which are also recurrent. Our findings theoretically justify that although SSMs variants exhibit good numerical performance in long-sequence modeling, simple SSMs also suffer from the "curse of memory".

State-space models (Siivola & Honkela, 2003), previously discussed in control theory, has been widely used to study the dynamics of complex systems. HIPPO (Gu et al., 2020), which is designed for the online compression of both continuous signals and discrete time series, generalizes the Legendre Memory Unit and provides a new memory update mechanism which achieves state-of-the-art for permutated MNIST. The subsequent variants, S4(Gu et al., 2021), GSS (Mehta et al., 2022), and S5 (Smith et al., 2023), have significantly enhanced empirical performance. Notably, they excel in the long-range arena (Tay et al., 2021), an area where transformers traditionally underperform. Contrary to the initial presumption, our investigations disclose that the ability to learn long-term memory is not derived from the linear RNN coupled with nonlinear layer-wise activations. Rather, our study underscores the benefits of stable reparameterization in both approximation and optimization.

Apart from the attempts in model design, several other approaches have been adopted in improving the long-term memory learning. Hardt et al. (2018) proposes the first polynomial guarantees to do linear system identification. It also points out gradient descent over (linear) RNN without reparameterization can blow up even with small learning rate. The training stability has also been discussed in Revay et al. (2020). Our paper shows that stable reparameterization has benefit in improving the optimization stability, which is important for training of large models (Wortsman et al., 2023).

## 6   CONCLUSION

In this paper, we study the intricacies of long-term memory learning in state-space models, specifically emphasizing the role of parameterization. We prove that state-space models without reparameterization fails to stably approximating targets that exhibit non-exponential decaying memory. Our analysis indicates this "curse of memory" phenomenon is caused by the recurrent weights converging to stability boundary. As an alternative, we introduce a class of stable reparameterization as a robust solution to this challenge, which also partially explains the performance of S4. We also explore the optimization advantages associated with stable reparameterization, especially concerning gradient-over-weight scale. Our results give the theoretical support to observed advantages of reparameterization in S4 and moreover give principled methods to design "best" reparameterization scheme in the optimization stability sense. This paper shows that stable reparameterization not only enables the stable approximation of any linear functionals with long-term memory but also enhances the optimization stability for general nonlinear functionals.

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

# A  THEORETICAL BACKGROUNDS

In this section, we collect the definitions for the theoretical statements.

## A.1  APPROXIMATION IN SOBOLEV NORM

**Definition A.1.** In sequence modeling as a nonlinear functional approximation problem, we consider the Sobolev norm of the functional sequence defined as follow:

$$\left\|\mathbf{H} - \widehat{\mathbf{H}}\right\|_{W^{1,\infty}} = \sup_t \left( \|H_t - \widehat{H}_t\|_\infty + \left\|\frac{dH_t}{dt} - \frac{d\widehat{H}_t}{dt}\right\|_\infty \right). \tag{13}$$

Here $\mathbf{H}$ is the target functional sequence to be approximated while the $\widehat{\mathbf{H}}$ is the model we use.

In particular, the nonlinear functional operator norm is given by:

$$\|H_t\|_\infty := \sup_{\mathbf{x} \neq \mathbf{0}} \frac{|H_t(\mathbf{x})|}{\|\mathbf{x}\|_\infty + 1} + |H(\mathbf{0})|. \tag{14}$$

As $\mathbf{H}(\mathbf{0}) = 0$, $\|H_t\|_\infty$ is reduced to $\sup_{\mathbf{x} \neq \mathbf{0}} \frac{|H_t(\mathbf{x})|}{\|\mathbf{x}\|_\infty + 1}$. If $\mathbf{H}$ is a linear functional, this definition is compatible with the common linear functional norm in Equation (19).

Here we check this operator norm in Equation (14) is indeed a norm. Without loss of generality, we will drop the time index for brevity.

1. Triangular inequality: For nonlinear functional $H_1$ and $H_2$,

$$\|H_1 + H_2\|_\infty := \sup_{\mathbf{x} \neq \mathbf{0}} \frac{|(H_1 + H_2)(\mathbf{x})|}{\|\mathbf{x}\|_\infty + 1} \tag{15}$$

$$\leq \sup_{\mathbf{x} \neq \mathbf{0}} \frac{|H_1(\mathbf{x})|}{\|\mathbf{x}\|_\infty + 1} + \sup_{\mathbf{x} \neq \mathbf{0}} \frac{|H_2(\mathbf{x})|}{\|\mathbf{x}\|_\infty + 1} = \|H_1\|_\infty + \|H_2\|_\infty. \tag{16}$$

   The inequality is by the property of supremum.

2. Absolute homogeneity: For any real constant $s$ and nonlinear functional $H$

$$\|sH\|_\infty := \sup_{\mathbf{x} \neq \mathbf{0}} \frac{|(sH)(\mathbf{x})|}{\|\mathbf{x}\|_\infty + 1} = |s| \sup_{\mathbf{x} \neq \mathbf{0}} \frac{|H(\mathbf{x})|}{\|\mathbf{x}\|_\infty + 1} = |s| \|H\|_\infty. \tag{17}$$

3. Positive definiteness: If $\|H\|_\infty = 0$, then for all non-zero inputs $\mathbf{x} \neq \mathbf{0}$ we have $H(\mathbf{x}) = 0$. As $H(\mathbf{0}) = 0$, then we know $H$ is a zero functional.

## A.2  RIESZ REPRESENTATION THEOREM FOR LINEAR FUNCTIONAL

**Theorem A.2** (Riesz-Markov-Kakutani representation theorem). *Assume $H : C_0(\mathbb{R}, \mathbb{R}^d) \mapsto \mathbb{R}$ is a linear and continuous functional. Then there exists a unique, vector-valued, regular, countably additive signed measure $\mu$ on $\mathbb{R}$ such that*

$$H(\mathbf{x}) = \int_\mathbb{R} x_s^\top d\mu(s) = \sum_{i=1}^d \int_\mathbb{R} x_{s,i} d\mu_i(s). \tag{18}$$

*In addition, we have the linear functional norm*

$$\|H\|_\infty := \sup_{\|\mathbf{x}\|_\mathcal{X} \leq 1} |H(\mathbf{x})| = \|\mu\|_1(\mathbb{R}) := \sum_i |\mu_i|(\mathbb{R}). \tag{19}$$

*In particular, this linear functional norm is compatible with the norm considered for nonlinear functionals in Equation (14).*

# B  PROOFS FOR THEOREMS AND LEMMAS

In Appendix B.1, we show that the nonlinear functionals defined by state-space models are point-wise continuous functionals at Heaviside inputs. In Appendix B.3, the proof for state-space models' exponential memory decaying memory property is given. In Appendix B.4, we prove the linear RNN with stable reparameterization can stably approximate any linear functional. The target is no longer limited to have an exponenitally decaying memory. The gradient norm estimate of the recurrent layer is included in Appendix B.5.

## B.1  PROOF FOR SSMS ARE POINT-WISE CONTINUOUS FUNCTIONALS

*Proof.* Let $\mathbf{x}$ be any fixed Heaviside input. Assume $\lim_{k\to\infty} \|\mathbf{x}_k - \mathbf{x}\|_\infty = 0$. Let $h_{k,t}$ and $h_t$ be the hidden state for inputs $\mathbf{x}_k$ and $\mathbf{x}$. Without loss of generality, assume $t > 0$. The following $|\cdot|$ refers to $p = \infty$ norm.

By definition of the hidden states dynamics and triangular inequality, since $\sigma(\cdot)$ is Lipschitz continuous

$$\frac{d|h_{k,t} - h_t|}{dt} = |\sigma(\Lambda h_{k,t} + U x_{k,t}) - \sigma(\Lambda h_t + U x_t)| \tag{20}$$

$$\leq L|\Lambda h_{k,t} + U x_{k,t} - \Lambda h_t - U x_t| \tag{21}$$

$$= L|\Lambda(h_{k,t} - h_t) + U(x_{k,t} - x_t)| \tag{22}$$

$$\leq L(|\Lambda||h_{k,t} - h_t| + |U||x_{k,t} - x_t|). \tag{23}$$

Here $L$ is the Lipschitz constant of activation $\sigma$. Apply the Grönwall inequality to the above inequality, we have:

$$|h_{k,t} - h_t| \leq \int_0^t e^{L|\Lambda|(t-s)} L|U| \, |x_{k,s} - x_s| ds. \tag{24}$$

As the inputs are bounded, by dominated convergence theorem we have right hand side converges to 0 therefore

$$\lim_{k\to\infty} |h_{k,t} - h_t| = 0, \quad \forall t. \tag{25}$$

Let $y_{k,t}$ and $y_t$ be the outputs for inputs $\mathbf{x}_k$ and $\mathbf{x}$. Therefore we show the point-wise convergence of $\frac{dH_t}{dt}$ at $\mathbf{x}$:

$$\lim_{k\to\infty} \left| \frac{dy_{k,t}}{dt} - \frac{dy_t}{dt} \right| = \lim_{k\to\infty} \left| c^\top (\frac{dh_{k,t}}{dt} - \frac{dh_t}{dt}) \right| \tag{26}$$

$$\leq \lim_{k\to\infty} |c| L (|\Lambda||h_{k,t} - h_t| + |U||x_{k,t} - x_t|) = 0. \tag{27}$$

$\square$

## B.2  POINT-WISE CONTINUITY LEADS TO DECAYING MEMORY

Here we give the proof of decaying memory based on the point-wise continuity of $\frac{dH_t}{dt}$ and boundedness and time-homogeneity of $\mathbf{H}$:

*Proof.*

$$\lim_{t\to\infty} \left| \frac{dH_t}{dt}(\mathbf{u}^x) \right| = \lim_{t\to\infty} \left| \frac{dH_0}{dt}(x \cdot \mathbf{1}_{\{s \geq -t\}}) \right| = \left| \frac{dH_0}{dt}(\mathbf{x}) \right| = 0.$$

The first equation comes from time-homogeneity. The second equation is derived from the point-wise continuity where input $\mathbf{x}$ means constant $x$ for all time $\mathbf{x} = x \cdot \mathbf{1}_{\{s \geq -\infty\}}$. The third equation is based on the boundedness and time-homogeneity as the output over constant input should be finite and constant $H_t(\mathbf{x}) = H_s(\mathbf{x})$ for all $s, t$. Therefore $|\frac{dH_0}{dt}(\mathbf{x})| = 0$. $\square$

### B.3 PROOF FOR THEOREM 3.1

The main idea of the proof is two-fold. First of all, we show that state-space models with strictly monotone activation is decaying memory in Lemma B.6. Next, the idea of analysing the memory functions through a transform from $[0, \infty)$ to $(0, 1]$ is similar to previous works (Li et al., 2020; 2022; Wang et al., 2023). The remainder of the proof follows a standard approach, as the derivatives of the hidden states follow the rules of linear dynamical systems when Heaviside inputs are considered.

*Proof.* Assume the inputs considered are uniformly bounded by $X_0$:

$$\|\mathbf{x}\|_\infty < X_0. \tag{28}$$

Define the derivative of hidden states for unperturbed model to be $v_{m,t} = \frac{dh_{m,t}}{dt}$. Similarly, $\tilde{v}_{m,t}$ is the derivative of hidden states for perturbed models $\tilde{v}_{m,t} = \frac{d\tilde{h}_{m,t}}{dt}$.

Since each perturbed model has a decaying memory and the target functional sequence $\mathbf{H}$ has a stable approximation, by Lemma B.6, we have

$$\lim_{t \to \infty} \tilde{v}_{m,t} = 0, \quad \forall m. \tag{29}$$

If the inputs are limited to Heaviside inputs, the derivative $\tilde{v}_{m,t}$ satisfies the following dynamics: Notice that the hidden state satisfies $h_t = 0, t \in (-\infty, 0]$,

$$\frac{d\tilde{v}_{m,t}}{dt} = \widetilde{\Lambda}_m \tilde{v}_{m,t}, \quad t \geq 0 \tag{30}$$

$$\tilde{v}_{m,0} = \widetilde{\Lambda}_m h_0 + \widetilde{U}_m x_0 + \tilde{b}_m = \widetilde{U}_m x_0 + \tilde{b}_m \tag{31}$$

$$\Rightarrow \tilde{v}_{m,t} = e^{\widetilde{\Lambda}_m t}(\widetilde{U}_m x_0 + \tilde{b}_m). \tag{32}$$

Notice that the perturbed initial conditions of the $\tilde{v}_{m,t}$ are uniformly (in $m$) bounded:

$$\tilde{V}_0 := \sup_m |\tilde{v}_{m,0}|_2 \tag{33}$$

$$= \sup_m |\widetilde{U}_m x_0 + \tilde{b}_m|_2 \tag{34}$$

$$\leq \sup_m |\widetilde{U}_m x_0 + \tilde{b}_m|_2 \tag{35}$$

$$\leq dX_0(\sup_m \|U_m\|_2 + \beta_0) + \sup_m \|b_m\|_2 + \beta_0 \tag{36}$$

$$< \infty \tag{37}$$

Here $d$ is the input sequence dimension.

Similarly, the unperturbed initial conditions satisfy:

$$V_0 := \sup_m |\tilde{v}_{m,0}|_2 \tag{38}$$

$$= \sup_m |U_m x_0 + b_m|_2 \tag{39}$$

$$\leq \sup_m |U_m x_0 + b_m|_2 \tag{40}$$

$$\leq dX_0 \sup_m \|U_m\|_2 + \sup_m \|b_m\|_2 \tag{41}$$

$$< \infty \tag{42}$$

Select a sequence of perturbed recurrent matrices $\{\widetilde{\Lambda}_{m,k}\}_{k=1}^\infty$ satisfying the following two properties:

1. $\widetilde{\Lambda}_{m,k}$ is Hyperbolic, which means the real part of the eigenvalues of the matrix are nonzero.

2. $\lim_{k \to \infty}(\widetilde{\Lambda}_{m,k} - \Lambda_m) = \beta_0 I_m$.

Moreover, by Lemma B.7, we know that each hyperbolic matrix $\widetilde{\Lambda}_{m,k}$ is Hurwitz as the system for $\tilde{v}_{m,t}$ is asymptotically stable.

$$\sup_m \max_{i \in [m]} (\lambda_i(\widetilde{\Lambda}_{m,k})) < 0. \tag{43}$$

This is the stability boundary for the state-space models under perturbations.

Therefore the original diagonal unperturbed recurrent weight matrix $\Lambda_m$ satisfies the following eigenvalue inequality **uniformly** in $m$. Since $\Lambda_m$ is diagonal:

$$\sup_m \max_{i \in [m]} (\lambda_i(\Lambda_m)) \leq -\beta_0. \tag{44}$$

Therefore the model memory decays exponentially uniformly

$$\mathcal{M}(\widehat{\mathbf{H}}_m)(t) := \sup_{X_0} \frac{1}{X_0 + 1} \left| \frac{d}{dt} \hat{y}_{m,t} \right| \tag{45}$$

$$= \sup_{X_0} \frac{1}{X_0 + 1} |c_m^\top [\sigma'(h_{m,t}) \circ v_{m,t}]| \tag{46}$$

$$\leq \sup_{X_0} \frac{1}{X_0 + 1} |c_m|_2 |\sigma'(h_{m,t}) \circ v_{m,t}|_2 \tag{47}$$

$$\leq \sup_{X_0} \frac{1}{X_0 + 1} |c_m|_2 \cdot \sup_z |\sigma'(z)| \cdot |e^{-\beta_0 t} v_{m,0}|_2 \tag{48}$$

$$\leq \sup_{X_0} \frac{1}{X_0 + 1} \left( \sup_m |c_m|_2 \cdot \sup_z |\sigma'(z)| \cdot V_0 \right) e^{-\beta_0 t} \tag{49}$$

$$\leq \sup_{X_0} \frac{1}{X_0 + 1} \left( \sup_m |c_m|_2 \cdot L_0 \cdot V_0 \right) e^{-\beta_0 t} \tag{50}$$

$$\leq \sup_{X_0} \left( \sup_m |c_m|_2 \cdot L_0 \right. \tag{51}$$

$$\left. \cdot \left( \frac{X_0}{X_0 + 1} d(\sup_m \|U_m\|_2) + \frac{1}{X_0 + 1}(\sup_m \|b_m\|_2) \right) \right) e^{-\beta_0 t} \tag{52}$$

$$\leq \left( \sup_m |c_m|_2 \cdot L_0 \left( d \sup_m \|U_m\|_2 + \sup_m \|b_m\|_2 \right) \right) e^{-\beta_0 t} \tag{53}$$

$$\leq (d + 1) L_0 \theta_{\max}^2 e^{-\beta_0 t} \tag{54}$$

The inequalities are based on vector norm properties, Lipschitz continuity of $\sigma(z)$ and uniform boundedness of unperturbed initial conditions. Therefore we know the model memories are uniformly decaying.

By Lemma B.8, the target $\mathbf{H}$ has an exponentially decaying memory as it is approximated by a sequence of models $\{\widehat{\mathbf{H}}_m\}_{m=1}^\infty$ with uniformly exponentially decaying memory. □

*Remark* B.1. When the approximation is unstable, we cannot have the real parts of the eigenvalues for recurrent weights bounded away from 0 in Equation (44). As the stability of linear RNNs requires the real parts (of the eigenvalues) to be negative, then the maximum of the real parts will converge to 0. This is the stability boundary of state-space models.

$$\lim_{m \to \infty} \max_{i \in [m]} (\lambda_i(\Lambda_m)) = 0^-. \tag{55}$$

*Remark* B.2. The uniform weights bound is necessary in the sense that: Since state-space models are universal approximators, they can approximate targets with long-term memories. However, if the target has an non-exponential decaying (e.g. polynomial decaying) memory, the weights bound of the approximation sequence will be exponential in the sequence length $T$.

$$\theta_{max}^2 \geq e^{\beta_0 T} \frac{\mathcal{M}(\mathbf{H})(T)}{(d + 1) L_0}. \tag{56}$$

This result indicates that scaling up SSMs without reparameterization is inefficient in learning sequence relationships with a large $T$ and long-term memory.

### B.4 PROOF FOR STABLE REPARAMETERIZATION ENABLES THE STABLE APPROXIMATION FOR LINEAR FUNCTIONAL

*Proof.* Let the target linear functional be $H_t(\mathbf{x}) = \int_{-\infty}^t \rho(t-s)x_s ds$. Here $\rho$ is an $L_1$ integrable function. We consider a simplified model setting with only parameters $c$ and $w$. Let $c_i, w_i$ be the unperturbed weights and $\tilde{w}_i$ be the perturbed recurrent weights. Similar to $\rho$ being $L_1$ integrable, we note that $\int_0^\infty |c_i e^{f(w_i)t}| dt = \frac{|c_i|}{|f(w_i)|}$. To have a sequence of well-defined model, we require they are uniformly (in $m$) absolutely integrable:

$$\sup_m \sum_{i=1}^m \frac{|c_i|}{|f(w_i)|} < \infty. \tag{57}$$

Based $|\tilde{w} - w|_2 \le \beta$. We know the approximation error is

$$E_m(\beta) = \sup_{|\tilde{w}-w|_2 \le \beta} \int_0^\infty \left| \sum_{i=1}^m c_i e^{f(\tilde{w}_i)t} - \rho(t) \right| dt \tag{58}$$

$$\le \sup_{|\tilde{w}-w|_2 \le \beta} \int_0^\infty \left| \sum_{i=1}^m c_i e^{f(w_i)t} - \rho(t) \right| dt \tag{59}$$

$$+ \sup_{|\tilde{w}-w|_2 \le \beta} \int_0^\infty \left| \sum_{i=1}^m c_i e^{f(\tilde{w}_i)t} - \sum_{i=1}^m c_i e^{f(w_i)t} \right| dt \tag{60}$$

$$= E_m(0) + \sup_{|\tilde{w}-w|_2 \le \beta} \int_0^\infty \sum_{i=1}^m |c_i| \left| e^{f(\tilde{w}_i)t} - e^{f(w_i)t} \right| dt \tag{61}$$

$$= E_m(0) + \sum_{i=1}^m |c_i| \sup_{|\tilde{w}-w|_2 \le \beta} \int_0^\infty \left| e^{f(\tilde{w}_i)t} - e^{f(w_i)t} \right| dt \tag{62}$$

$$\le E_m(0) + \sum_{i=1}^m |c_i| \sup_{|\tilde{w}_i-w_i|_2 \le \beta} \int_0^\infty \left| e^{f(\tilde{w}_i)t} - e^{f(w_i)t} \right| dt \tag{63}$$

$$\le E_m(0) + \sum_{i=1}^m |c_i| \frac{g(\beta)}{|f(w_i)|} \tag{64}$$

$$= E_m(0) + \sum_{i=1}^m \frac{|c_i|}{|f(w_i)|} g(\beta). \tag{65}$$

The first inequality is the triangular inequality. The second inequality comes from the fact that $|\tilde{w}_i - w_i| \le |\tilde{w} - w|_2 \le \beta$. The third inequality is achieved via the property of stable reparameterization: For some continuous function $g(\beta) : [0, \infty) \to [0, \infty), g(0) = 0$:

$$\sup_w \left[ |f(w)| \sup_{|\tilde{w}-w| \le \beta} \int_0^\infty \left| e^{f(\tilde{w})t} - e^{f(w)t} \right| dt \right] \le g(\beta). \tag{66}$$

By definition of stable approximation, we know $\lim_{m\to\infty} E_m(0) = 0$. Also according to the requiement of the stable approximation in Equation (57), we have

$$\lim_{\beta \to 0} E(\beta) = \lim_{\beta \to 0} \lim_{m \to \infty} E_m(\beta) \tag{67}$$

$$\le \lim_{\beta \to 0} \lim_{m \to \infty} E_m(0) + \left( \sup_m \sum_{i=1}^m \frac{|c_i|}{|f(w_i)|} \right) * \lim_{\beta \to 0} g(\beta) \tag{68}$$

$$= 0 + 0 = 0 = E(0). \tag{69}$$

Since $g(\beta)$ is continuous, it can be seen $E(\beta)$ is continuous:

$$E(\beta) \le \left( \sup_m \sum_{i=1}^m \frac{|c_i|}{|f(w_i)|} \right) g(\beta). \tag{70}$$

□

*Remark* B.3. **Here we verify the reparameterization methods satisfy the definition of stable reparameterization.**

For exponential reparameterization $f(w) = -e^w, w \in \mathbb{R}$:

$$\sup_{|\tilde{w}-w|\leq\beta} \int_0^\infty \left| e^{f(\tilde{w})t} - e^{f(w)t} \right| dt = \frac{e^\beta - 1}{|f(w)|}. \tag{71}$$

For softplus reparameterization $f(w) = -\log(1 + e^w), w \in \mathbb{R}$: Notice that $\exp(-\beta)\log(1 + \exp(w)) \leq \sup_{|\tilde{w}-w|\leq\beta} \log(1 + \exp(\tilde{w})) \leq \exp(\beta)\log(1 + \exp(w))$,

$$\sup_{|\tilde{w}-w|\leq\beta} \int_0^\infty \left| e^{f(\tilde{w})t} - e^{f(w)t} \right| dt \leq \frac{e^\beta - 1}{|f(w)|}. \tag{72}$$

For "best" reparameterization $f(w) = -\frac{1}{aw^2+b}, w \in \mathbb{R}, a, b > 0$: Without loss of generality, let $w \geq 0$

$$\sup_{|\tilde{w}-w|\leq\beta} \int_0^\infty \left| e^{f(\tilde{w})t} - e^{f(w)t} \right| dt = |a(w+\beta)^2 - aw^2| \tag{73}$$

$$\leq \frac{\frac{a(\beta^2+2\beta w)}{aw^2+b}}{|f(w)|} \tag{74}$$

$$\leq \frac{\frac{a(\beta^2+2\beta w)}{b}}{|f(w)|}. \tag{75}$$

Here $g(\beta) = \frac{a(\beta^2+2\beta w)}{b}$. The famous Müntz–Szász theorem indicates that selecting any non-zero constant $a$ does not affect the universality of linear RNN.

While for the case without reparameterization $f(w) = w, w < 0$: For $0 \leq \beta < -w$,

$$\sup_{|\tilde{w}-w|\leq\beta} \int_0^\infty \left| e^{f(\tilde{w})t} - e^{f(w)t} \right| dt = \frac{\beta}{(-w-\beta)(-w)} = \frac{\beta}{(-w-\beta)|f(w)|}, \tag{76}$$

Here $\lim_{w\to-\beta} \sup_w \frac{\beta}{-w-\beta} = \infty$, therefore the direct parameterization is not a stable reparameterization.

## B.5 PROOF FOR THEOREM 3.4

*Proof.* For any $1 \leq j \leq m$, assume the loss function we used is the $L_\infty$ norm: Loss $= \sup_t \|H_t - \widehat{H}_{m,t}\|_\infty$. Notice that by time-homogeneity, Loss $= \|H_t - \widehat{H}_{m,t}\|_\infty$ for any $t$. This loss function is larger than the common mean squared error, which is usually chosen in practice for the smoothness reason.

$$\left|\frac{\partial \text{Loss}}{\partial w_j}\right| = \left|\frac{\partial \|H_t - \widehat{H}_{m,t}\|_\infty}{\partial w_j}\right| \tag{77}$$

$$= \left|\frac{\partial \sup_{\|\mathbf{x}\|_\infty \leq 1} |H_t(\mathbf{x}) - \widehat{H}_{m,t}(\mathbf{x})|}{\partial w_j}\right| \tag{78}$$

$$= \left|\frac{\partial \sup_{\|\mathbf{x}\|_\infty \leq 1} |\int_{-\infty}^t (\rho(t-s) - \sum_{i=1}^m c_i e^{-f(w_i)(t-s)}) x_s ds|}{\partial w_j}\right| \tag{79}$$

$$= \left|\frac{\partial \int_{-\infty}^t |\rho(t-s) - \sum_{i=1}^m c_i e^{-f(w_i)(t-s)}| ds}{\partial w_j}\right| \tag{80}$$

$$= \left|\frac{\partial \int_{-\infty}^t |(\rho(t-s) - \sum_{i\neq j} c_i e^{-f(w_i)(t-s)}) - c_j e^{-f(w_j)(t-s)}| ds}{\partial w_j}\right| \tag{81}$$

$$= \left|\frac{\partial \int_0^\infty |(\rho(s) - \sum_{i\neq j} c_i e^{-f(w_i)s}) - c_j e^{-f(w_j)s}| ds}{\partial w_j}\right| \tag{82}$$

$$\leq \int_0^\infty \left|\frac{\partial |(\rho(s) - \sum_{i\neq j} c_i e^{-f(w_i)s}) - c_j e^{-f(w_j)s}|}{\partial w_j}\right| ds \tag{83}$$

$$\leq \int_0^\infty \left|\frac{\partial |c_j e^{-f(w_j)s}|}{\partial w_j}\right| ds \tag{84}$$

The first equality is the definition of the loss function. The second equality equality comes from the definition of the linear functional norm. The third equality expand the linear functional and linear RNNs into the convolution form. The fourth equality utilize the fact that we can manually select $x_t$'s sign to achieve the maximum value. The fifth equality is separating the term in dependent of variable $w_j$. The sixth equality is change of variable from $t - s$ to $s$. The inequality is triangular inequality. The last equality is dropping the term independent of variable $w_j$.

$$\left|\frac{\partial \text{Loss}}{\partial w_j}\right| \leq \int_0^\infty \left|\frac{\partial |c_j e^{-f(w_j)s}|}{\partial w_j}\right| ds \tag{85}$$

$$= |c_j f'(w_j)| \int_0^\infty e^{-f(w_j)s} s \, ds \tag{86}$$

$$= \left|c_j \frac{f'(w_j)}{f(w_j)}\right| \int_0^\infty e^{-f(w_j)s} ds \tag{87}$$

$$= \left|c_j \frac{f'(w_j)}{f(w_j)^2}\right| (1 - \lim_{s\to\infty} e^{-f(w_j)s}) = \left|c_j \frac{f'(w_j)}{f(w_j)^2}\right|. \tag{88}$$

The first equality is evaluating the derivative. The second equality is extracting $|f'(w)|$ from integral. The third equality is doing the integration by parts.

In particular, notice that $c_j$ is a constant independent of the recurrent weight parameterization $f$:

$$\widehat{H}_{m,t}(\mathbf{x}) = \int_{-\infty}^t \sum_{i=1}^m c_i e^{-f(w_i)(t-s)} x_s ds. \tag{89}$$

Therefore $c_j$ is a parameterization indepndent value, we will denote it by $C_{\mathbf{H},\widehat{\mathbf{H}}_m}$.

Moreover, in the discrete setting, assume $h_{k+1} = f(w) \circ h_k + Ux_k$,

$$\left| \frac{\partial \text{Loss}}{\partial w_j} \right| \leq \sum_{k=0}^{\infty} \left| \frac{\partial |c_j f(w_j)^k|}{\partial w_j} \right| ds \tag{90}$$

$$= |c_j f'(w_j)| \sum_{k=1}^{\infty} k f(w_j)^{k-1} \tag{91}$$

$$= |c_j f'(w_j)| \left( \sum_{k=1}^{\infty} f(w_j)^{k-1} \right)^2 \tag{92}$$

$$= \left| c_j \frac{f'(w_j)}{(1 - f(w_j))^2} \right|. \tag{93}$$

So the gradient norm is bounded by

$$\left| \frac{\partial \text{Loss}}{\partial w_j} \right| = \frac{|c_j f'(w_j)|}{(1 - f(w_j))^2}. \tag{94}$$

$\square$

**Nonlinear functionals**   Now we show the generalization into the nonlinear functional: Consider the Volterra Series representation of the nonlinear functional.

**Theorem B.4** ((Boyd et al., 1984)). *For any continuous time-invariant system with $x(t)$ as input and $y(t)$ as output can be expanded in the Volterra series as follow*

$$y(t) = h_0 + \sum_{n=1}^{N} \int_0^t \cdots \int_0^t h_n(\tau_1, \ldots, \tau_n) \prod_{j=1}^{n} x(t - \tau_j) d\tau_j. \tag{95}$$

*Here $N$ is the series' order. Linear functional is an order-1 Volterra series.*

For simplicity, we will only discuss the case for $N = 2$. When we take the Hyena approach (Poli et al., 2023) and approximate the order-2 kernel $h_2(\tau_1, \tau_2)$ with its rank-1 approximation:

$$h_2(\tau_1, \tau_2) = h_{2,1}(\tau_1) h_{2,2}(\tau_2). \tag{96}$$

Here $h_{2,1}$ and $h_{2,2}$ are again order-1 kernel which can be approximated with linear RNN's kernel. In other words, the same gradient bound also holds for general nonlinear functional with the following form:

$$G_f(w) := \left| \frac{\partial E}{\partial w} \right| = C_{\mathbf{H}, \widehat{\mathbf{H}}_m} \frac{|f'(w)|}{f(w)^2}. \tag{97}$$

And the discrete version is

$$G_f^D(w) := \left| \frac{\partial E}{\partial w} \right| = C_{\mathbf{H}, \widehat{\mathbf{H}}_m} \frac{|f'(w)|}{(1 - f(w))^2}. \tag{98}$$

### B.6   LEMMAS

**Lemma B.5.** *If the activation $\sigma(\cdot)$ is bounded, strictly increasing, continuously differentiable function over $\mathbb{R}$. Then for all $C > 0$, there exists $\epsilon_C$ such that $\forall |z| \leq C_\epsilon$, $|\sigma'(z)| \geq \epsilon_C$.*

*Proof.* Since $\sigma(\cdot)$ is monotonically increasing, therefore $\sigma'(\cdot) > 0, \forall z \geq 0$. Notice that $\sigma'(\cdot)$ is continuous, for any $C > 0$, we know $\frac{1}{2} \min_{|z| \leq C} \sigma'(z) > 0$. Define $\epsilon_C := \frac{1}{2} \min_{|z| \leq C} \sigma'(z) > 0$, it can be seen the target statement is satisfied. $\square$

**Lemma B.6.** *Assume the target functional sequence has a $\beta_0$-stable approximation and the perturbed model has a decaying memory, we show that $\tilde{v}_{m,t} \to 0$ for all $m$.*

*Proof.* For any $m$, fix $\widetilde{\Lambda}_m$ and $\widetilde{U}_m$. Since the perturbed model has a decaying memory,

$$\lim_{t\to\infty}\left|\frac{d}{dt}\widetilde{H}_m(\mathbf{u}^x)\right| = \lim_{t\to\infty}\left|c^\top(\sigma'(\tilde{h}_{m,t})\circ\frac{d\tilde{h}_{m,t}}{dt})\right| = \lim_{t\to\infty}\left|c^\top(\sigma'(\tilde{h}_{m,t})\circ\tilde{v}_{m,t})\right| = 0. \quad (99)$$

By linear algebra, there exist $m$ vectors $\{\Delta c_i\}_{i=1}^m, |\Delta c_i|_\infty < \beta$ such that $c_m + \Delta c_1, \dots, c_m + \Delta c_m$ form a basis of $\mathbb{R}^m$. We can then decompose any vector $u$ into

$$u = k_1(c_m + \Delta c_1) + \cdots + k_m(c_m + \Delta c_m). \quad (100)$$

Take the inner product of $u$ and $\tilde{v}_{m,t}$, we have

$$\lim_{t\to\infty} u^\top(\sigma'(\tilde{h}_{m,t})\circ\tilde{v}_{m,t}) = \sum_{i=1}^m k_i \lim_{t\to\infty}(c_m + \Delta c_i)^\top(\sigma'(\tilde{h}_{m,t})\circ\tilde{v}_{m,t}) = 0 \quad (101)$$

As the above result holds for any vector $u$, we get

$$\lim_{t\to\infty}\left|\sigma'(\tilde{h}_{m,t})\circ\tilde{v}_{m,t}\right|_\infty = 0. \quad (102)$$

As required in Equation (4), the hidden states are uniformly (in $m$) bounded over bounded input sequence. There exists constant $C_0 > 0$ such that

$$\sup_{m,t}|h_{m,t}|_\infty < C_0. \quad (103)$$

Since $\sigma$ is continuously differentiable and strictly increasing, by Lemma B.5, there exists $\epsilon_{C_0} > 0$ such that

$$|\sigma'(z)| > \epsilon_{C_0}, \quad \forall|z| \le C_0. \quad (104)$$

Therefore

$$\sup_t\left|\sigma'(\tilde{h}_{m,t})\right|_\infty > \epsilon_{C_0}. \quad (105)$$

We get

$$\lim_{t\to\infty}|\tilde{v}_{m,t}|_\infty = 0. \quad (106)$$

$\square$

**Lemma B.7.** *Consider a dynamical system with the following dynamics:* $h_0 = 0$

$$\begin{aligned}\frac{dv_t}{dt} &= \Lambda v_t, \\ v_0 &= \Lambda h_0 + \widetilde{U}x_0 + \tilde{b} = \widetilde{U}x_0 + \tilde{b}.\end{aligned} \quad (107)$$

*If $\Lambda \in \mathbb{R}^{m\times m}$ is diagonal, hyperbolic and the system in Equation (107) is satisfies $\lim_{t\to\infty} v_t = 0$ over any bounded Heaviside input $\mathbf{u}^{x_0}, |x_0|_\infty < \infty$, then the matrix $\Lambda$ is Hurwitz.*

*Proof.* By integration we have the following explicit form:

$$v_t = e^{\Lambda t}v_0 = e^{\Lambda t}(\widetilde{U}x_0 + \tilde{b}). \quad (108)$$

The stability requires $\lim_{t\to\infty}|v_t| = 0$ for all inputs $v_0 = \widetilde{U}x_0 + \tilde{b}$. Notice that with perturbation from $\tilde{U}$ and $\tilde{b}$, the set of initial points $\{v_0\}$ is m-dimensional. Therefore the matrix $\Lambda$ is Hurwitz in the sense that all eigenvalues' real parts are negative. $\square$

**Lemma B.8.** *Consider a continuous function $f : [0,\infty) \to \mathbb{R}$, assume it can be approximated by a sequence of continuous functions $\{f_m\}_{m=1}^\infty$ universally:*

$$\lim_{m\to\infty}\sup_{t\ge0}|f(t) - f_m(t)| = 0. \quad (109)$$

*Assume the approximators $f_m$ are uniformly exponentially decaying with the same $\beta_0 > 0$:*

$$\lim_{t\to\infty}\sup_{m\in\mathbb{N}_+} e^{\beta_0 t}|f_m(t)| \to 0. \quad (110)$$

*Then the function $f$ is also decaying exponentially:*

$$\lim_{t\to\infty} e^{\beta t}|f(t)| \to 0, \quad \forall 0 < \beta < \beta_0. \quad (111)$$

The proof is the same as Lemma A.11 from (Wang et al., 2023). For completeness purpose, we attach the proof here:

*Proof.* Given a function $f \in C([0, \infty))$, we consider the transformation $\mathcal{T}f : [0, 1] \to \mathbb{R}$ defined as:

$$(\mathcal{T}f)(s) = \begin{cases} 0, & s = 0 \\ \frac{f(-\frac{\log s}{\beta_0})}{s}, & s \in (0, 1]. \end{cases} \tag{112}$$

Under the change of variables $s = e^{-\beta_0 t}$, we have:

$$f(t) = e^{-\beta_0 t}(\mathcal{T}f)(e^{-\beta_0 t}), \quad t \geq 0. \tag{113}$$

According to uniformly exponentially decaying assumptions on $f_m$:

$$\lim_{s \to 0^+} (\mathcal{T}f_m)(s) = \lim_{t \to \infty} \frac{f_m(t)}{e^{-\beta_0 t}} = \lim_{t \to \infty} e^{\beta_0 t} f_m(t) = 0, \tag{114}$$

which implies $\mathcal{T}f_m \in C([0, 1])$.

For any $\beta < \beta_0$, let $\delta = \beta_0 - \beta > 0$. Next we have the following estimate

$$\sup_{s \in [0,1]} |(\mathcal{T}f_{m_1})(s) - (\mathcal{T}f_{m_2})(s)| \tag{115}$$

$$= \sup_{t \geq 0} \left| \frac{f_{m_1}(t)}{e^{-\beta t}} - \frac{f_{m_2}(t)}{e^{-\beta t}} \right| \tag{116}$$

$$\leq \max \left\{ \sup_{0 \leq t \leq T_0} \left| \frac{f_{m_1}(t)}{e^{-\beta t}} - \frac{f_{m_2}(t)}{e^{-\beta t}} \right|, C_0 e^{-\delta T_0} \right\} \tag{117}$$

$$\leq \max \left\{ e^{\beta T_0} \sup_{0 \leq t \leq T_0} |f_{m_1}(t) - f_{m_2}(t)|, C_0 e^{-\delta T_0} \right\} \tag{118}$$

where $C_0$ is a constant uniform in $m$.

For any $\epsilon > 0$, take $T_0 = -\frac{\ln(\frac{\epsilon}{C_0})}{\delta}$, we have $C_0 e^{-\delta T_0} \leq \epsilon$. For sufficiently large $M$ which depends on $\epsilon$ and $T_0$, by universal approximation (Equation (109)), we have $\forall m_1, m_2 \geq M$,

$$\sup_{0 \leq t \leq T_0} |f_{m_1}(t) - f_{m_2}(t)| \leq e^{-\beta T_0} \epsilon, \tag{119}$$

$$e^{\beta T_0} \sup_{0 \leq t \leq T_0} |f_{m_1}(t) - f_{m_2}(t)| \leq \epsilon. \tag{120}$$

Therefore, $\{f_m\}$ is a Cauchy sequence in $C([0, \infty))$.

Since $\{f_m\}$ is a Cauchy sequence in $C([0, \infty))$ equipped with the sup-norm, using the above estimate we can have $\{\mathcal{T}f_m\}$ is a Cauchy sequence in $C([0, 1])$ equipped with the sup-norm. By the completeness of $C([0, 1])$, there exists $f^* \in C([0, 1])$ with $f^*(0) = 0$ such that

$$\lim_{m \to \infty} \sup_{s \in [0,1]} |(\mathcal{T}f_m)(s) - f^*(s)| = 0. \tag{121}$$

Given any $s > 0$, we have

$$f^*(s) = \lim_{m \to \infty} (\mathcal{T}f_m)(s) = (\mathcal{T}f)(s), \tag{122}$$

hence

$$\lim_{t \to \infty} e^{\beta t} f(t) = \lim_{s \to 0^+} (\mathcal{T}f)(s) = f^*(0) = 0. \tag{123}$$

$\square$

## C  NUMERICAL DETAILS

In this section, the details of numerical experiments are provided for the completeness and reproducibility.

## C.1 SYNTHETIC TASK

We conduct the approximation of linear functional with linear RNNs in the one-dimensional input and one-dimensional output case. The synthetic linear functional is constructed with the polynomial decaying memory function is $\rho(t) = \frac{1}{(t+1)^{1.1}}$. Sequence length is 100. Total number of synthetic samples is 153600. The learning rate used is 0.01 and the batch size is 512.

The perturbation list $\beta \in [0, 10^{-3}, 10^{-3} * 2^{1/2}, 10^{-3} * 2^{2/2}, \ldots, 10^{-3} * 2^{20/2}]$. Each evaluation of the perturbed error is sampled with 30 different weight perturbations to reduce the variance.

## C.2 LANGUAGE MODELS

The language modeling is done over WikiText-103 dataset (Merity et al., 2016). The model we used is based on the Hyena architecture with simple real-weights state-space models as the mixer (Poli et al., 2023; Smith et al., 2023). The batch size is 16, total steps 115200 (around 16 epochs), warmup steps 1000. The optimizer used is AdamW and the weight decay coefficient is 0.25. The learning rate for the recurrent layer is 0.004 while the learning rate for other layers are 0.005.

# D ADDITIONAL NUMERICAL RESULTS

## D.1 ADDITIONAL NUMERICAL RESULTS FOR LANGUAGE MODELS

In the main paper, we provide the training loss curve for learning rate = 0.005 as the stability of "best" discrete-time parameterization $f(w) = 1 - \frac{1}{w^2+0.5}$ is mostly significant as the learning rate is large. In Figure 6, we further provide the results for other learning rates (lr = 0.001, 0.002, 0.010). Despite the final loss not being optimal for the "best" reparameterization, it is observed that the training process exhibits enhanced stability compared to other parameterization methods.

| Reparameterizations | Train ppl | Train loss | Test ppl | Test loss |
|:---:|:---:|:---:|:---:|:---:|
| "Best" | 17.182 | 2.844 | 20.811 | 3.035 |
| Exp(S5) | 15.721 | 2.755 | 20.218 | 3.007 |
| Softplus | **14.570** | **2.679** | **20.136** | **3.003** |
| Direct | 18.916 | 2.940 | 28.167 | 3.338 |

Table 2: Train/test perplexity and loss for the language modelling over wikitext103, lr=0.005. The models with stable reparameterizations are all better than the models without reparameterizations.

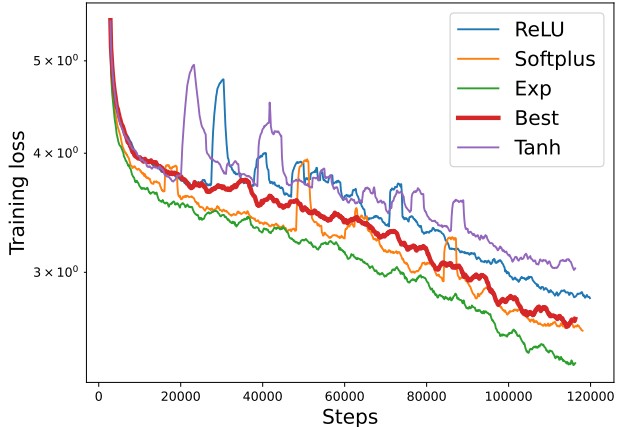

(a) lr=0.001, "best" reparameterization is not optimal in loss but no large fluctuation. Exponential parameterisation is also stable at lr=0.001

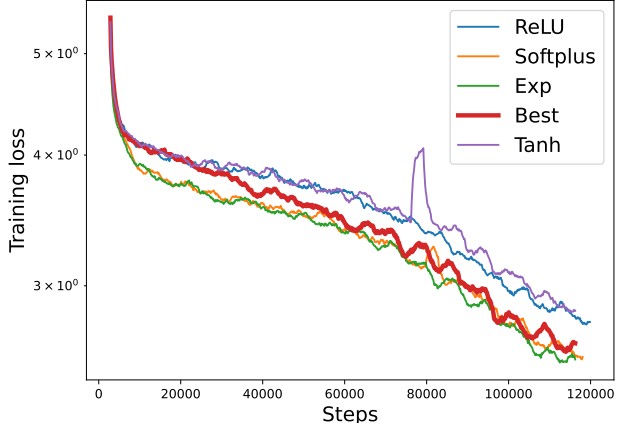

(b) lr=0.002, "best" reparameterization is also not optimal, but the final loss is comparable against Exp and Softplus

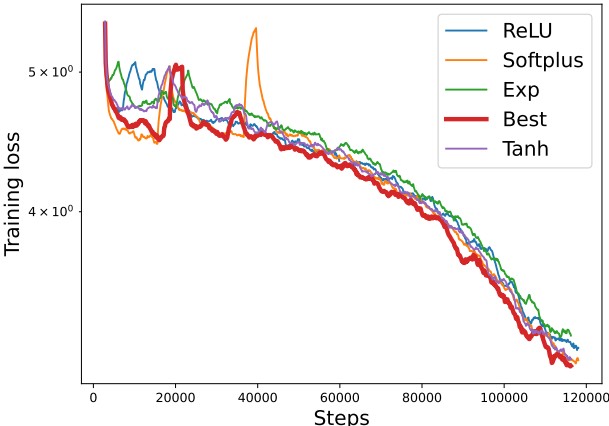

(c) lr=0.01, "best" reparameterization achieve the smallest loss

Figure 6: The stability advantage of "best" reparameterization (red line) is usually better when the learning rate is larger.

### D.2 ON THE STABILITY OF "BEST" REPARAMETERIZATION

The previous experiment on Wikitext language modelling shows the performance of stable reparameterization over the unstable cases. We further verify the optimization stability of "best" reparameterization in the following extreme setting. We construct a large scale language model with 3B parameters and train with larger learning rate (lr=0.01). As can be seen in the following table, the only convergent model is the model with "best" reparameterization. We emphasize that the only difference between these models are the parameterization schemes for recurrent weights. Therefore the best reparameterization is the most **stable** parameterization. (We repeats the experiments with different seeds for three times.)

|  | "Best" | Exp | Softplus | Direct |
|---|---|---|---|---|
| Convergent / total experiments | 3/3 | 0/3 | 0/3 | 0/3 |

Table 3: Experiment to the stability of "best" reparameterization over lr = 0.01. All other reparameterizations diverged within 100 steps while the "best" reparameterizations can be used to train the model.

### D.3 ADDITIONAL NUMERICAL RESULTS FOR ASSOCIATIVE RECALLS

In this section, we study the performance of of different stable reparameterizations over the extremely long sequences (up to 131k). It can be seen in Table 4 that stable parameterizations are better than the case without reparameterization and simple clipping. The advantage is more significant when the sequence length is longer. The models are trained under the exactly same hyperparameters.

| Reparameterizations | Train acc | Test acc | Train acc | Test acc |
|---|---|---|---|---|
| **"Best"** | **57.95** | **99.8** | **53.57** | **100** |
| Exp(S5) | 54.55 | **99.8** | **53.57** | **100** |
| Clip | 50.0 | 76.6 | 13.91 | 9.4 |
| Direct | 43.18 | 67.0 | 16.59 | 5.6 |

Table 4: Comparison of parameterizations on associative recalls. The first two columns are the train and test accuracy over **sequence length 20**, vocabulary size 10, while the second two columns are the train and test accuracy over **sequence length 131k** and vocabulary size 30.

### D.4 ADDITIONAL NUMERICAL RESULTS FOR IMAGE CLASSIFICATIONS

In this section, we study the effects of stable reparameterization over the image classfication tasks. For the fairness of comparison, we set the hyperparameters and initialization schemes to be exactly the same. All models are trained for 10 epochs. While the direct parameterization of the recurrent weights will cause the divergence, the stable reparameterization enables the learning of sequential MNIST in Table 5 and sequential CIFAR10 in Table 6.

## E GRAPHICAL DEMONSTRATION OF STATE-SPACE MODELS AS STACK OF EQUATION (2)

Here we show that the Equation (2) correspond to the practical instantiation of SSM-based models in the following sense: As shown in Figure 7, any practical instantiation of SSM-based models

|  | Train acc | Train loss | Test acc | Test loss |
|---|---|---|---|---|
| **"Best"** | **99.31(0.0264)** | **0.02269(0.00121)** | **99.3(0.153)** | **0.02501(0.00519)** |
| Exp | 99.26(0.241) | 0.02305(0.000694) | 99.21 (0.0321) | 0.02546(0.00118) |
| Softplus | 99.14(0.036) | 0.02337(0.00132) | 99.19 (0.0458) | 0.02682(0.000769) |
| Direct | 9.84(0.0753) | Diverged | 10.09 (0.742) | Diverged |

Table 5: Comparison of parameterizations on image classification over MNIST. The best reparameterization scheme $f(w) = 1 - \frac{1}{w^2+0.5}$ comes with the best performance over the average of three repeats. The standard deviation is included in the parenthesis.

| Reparameterizations | Train acc | Train loss | Test acc | Test loss |
|---|---|---|---|---|
| **"Best"** | **61.97(0.500)** | **1.059(0.0155)** | **66.41(0.156)** | **0.9430(0.00248)** |
| Exp | 61.77 (0.690) | 1.062(0.0141) | 65.87(0.136) | 0.9487(0.00983) |
| Softplus | 61.53 (0.716) | 1.0683(0.0142) | 65.84(0.05) | 0.9575(0.00189) |
| Direct | 9.988 (0.032) | Diverged | 9.736(8.533E-05) | Diverged |

Table 6: Comparison of parameterizations on image classification over CIFAR10. The best reparameterization scheme $f(w) = 1 - \frac{1}{w^2+0.5}$ comes with the best performance over the average of three repeats. The standard deviation is included in the parenthesis.

can be implemented as a stack of Equation (2). The pointwise shallow MLP can be realized with state-space model layer with layer-wise nonlinearity by setting recurrent weights $W$ to be 0.

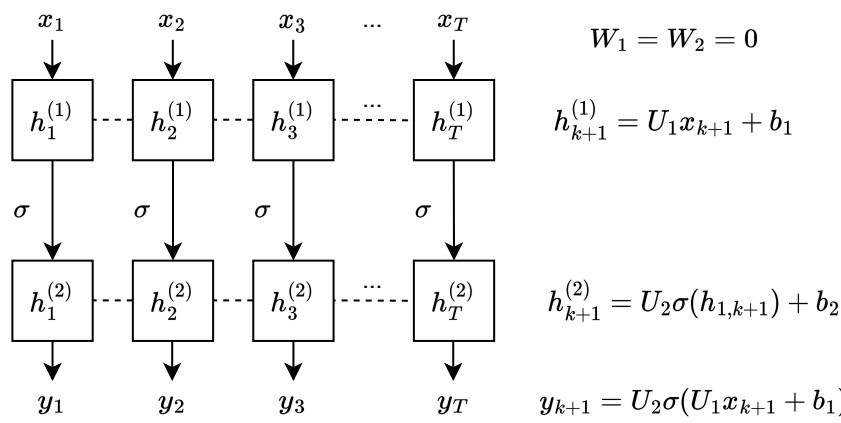

Figure 7: MLP can be realized by two-layer state-space models. The superscript indicates the layers while the subscript indicates the time index. It can be seen the MLP is equivalent to having zero recurrent weights $W_1 = W_2 = 0$.

## F   MOTIVATION FOR THE GRADIENT-TO-WEIGHT LIPSCHITZ CRITERION

Here we discuss the motivation for adopting the gradient-over-weight boundedness as a criterion. First of all, the "best" reparameterization is proposed to further improve the optimization stability across different memory patterns. The criterion "gradient is Lipschitz to the weight" is a necessary condition for the stability in the following sense:

1. Consider functions $f(x) = x^4$, the gradient function $g(x) = 4x^3$ does not have a global Lipschitz coefficient for all input values $x$. Therefore for any fixed positive learning rate $\eta$, there exists an initial point $x_0$ (for example $x_0 = \frac{1}{2\eta} + 1$) such that the convergence cannot

be achieved via the gradient descent step

$$x_{k+1} = x_k - \eta g(x). \tag{124}$$

2. Consider functions $f(x) = x^2$, the gradient function $g(x) = 2x$ is associated with a Lipschitz constant $L = 2$. Then the same gradient descent step converges for any $\eta \leq \frac{1}{2}$ in Equation (124).

3. As can be seen in the above two examples, **the criterion "gradient is Lipschitz to the weight" is associated with the convergence under large learning rate.** As the use of larger learning rate is usually associated with faster convergence (Smith & Topin, 2019), smaller generalization errors (Li et al., 2019), we believe the Lipschitz criterion is a suitable stability criterion for the measure of optimization stability.

4. The gradient-to-weight ratio evaluated in Figure 4(a) is a numerical verification of our Theorem 3.4. The gradients of stable reparameterizations are less susceptible to the well-known issue of exploding or vanishing gradients (Bengio et al., 1994; Hochreiter, 1998).

## G  SOLUTION OF "BEST" PARAMETERIZATION BASED ON ODE

Here we give a brief proof to show the "best" parameterization is the optimal "hypernetwork" in the sense of gradient-over-weight Lipschitz continuity. Further study of the optimization of hypernetworks for other criterions are left for future works.

*Proof.* Assume for some constant $L$, $G_f(w) \equiv L|w|$ for all $w$,

$$G_f(w) = C_{\mathbf{H}, \widehat{\mathbf{H}}_m} \frac{f'(w)}{f(w)^2} = Lw, \tag{125}$$

$$\frac{d(\ln(f(w)))}{dw} = aw, \quad a = \frac{L}{C_{\mathbf{H}, \widehat{\mathbf{H}}_m}} \tag{126}$$

$$\ln(f(w)) = aw^2 + b. \tag{127}$$

The second and third equation are achieved by integrating the function $\frac{f'(w)}{f(w)^2}$. Here $a, b \in \mathbb{R}$. Therefore the "best" reparameterization under the assumption of the Lipschitz property of gradient is characterized by the function with two degrees of freedom: By stability requirement $f(w) \leq 0$

$$f(w) = \frac{1}{aw^2 + b}, \quad a \neq 0.$$

$\square$

