# OpenReview forum: "StableSSM: Alleviating the Curse of Memory in State-space Models through Stable Reparameterization"
_ICLR.cc/2024/Conference — Submitted to ICLR 2024_

### Official Review · Reviewer_4Tez · 2023-10-31

**Soundness:** 2 fair
**Presentation:** 3 good
**Contribution:** 3 good
**Rating:** 5
**Confidence:** 2

**Summary:**

The paper is about alleviating the "curse of memeory" in sequence modeling. Authors treat the training of a state-space model (SSM or linear RNN) as an estimation of regularized linear functional $\mathbf{H}$, and proves that the normal SSM with stable approximation shows exponential decay and thus cannot estimate functions with non-exponential decay. Then, it is proved that using reparameterization such as softplus can alleviate this problem, and also suggest optimal reparameterization operator for the given task.

**Strengths:**

The main theorems about the curse of memory and the necessity of reparameterization is thoroughly supported with assumptions and resulting proofs.

**Weaknesses:**

1. The analysis on the curse of memeory is limited to the simple state space model approximating linear decay. Reparameterization technique may not be applicable for sequential models with more complex structures.

2. It is not clear that why the "best" reparameterization should satisfy that the gradient is Lipshitz to the weight.

**Questions:**

1. As the numerical examples mainly show the results on gradient to weight ratio, how can you justify that this quantity is related with better training?

2. Can this analysis be extended to more complex models such as multi-layer RNN or transformer-based RNN?

---

> ### Author Response · Authors · 2023-11-22
>
> We thank the reviewer for their thoughtful comments and constructive feedback.
>
> Weaknesses:
> 1. `The analysis on the curse of memeory is limited to the simple state space model approximating linear decay. Reparameterization technique may not be applicable for sequential models with more complex structures.`
>
>     The reparameterization technique can also be applied to sequential models with more complex structures in the following sense:
>     In multi-layer state-space models, we implement the analysis outlined in Theorem 3.4 on a layer-by-layer basis.
>     When examining the gradient of recurrent weights, denoted as $\lambda=f(w)$, the same bounds on gradient norm are applicable even for more complex structures.
>         $$G_{f}(w) := \left | \frac{\partial \textrm{Loss}}{\partial w} \right | \leq C_{\mathbf{H}, \widehat{\mathbf{H}}_m} \frac{|f'(w)|}{f(w)^2}.$$
>     The coefficient $C$ is a target-dependent and model-dependent coefficient, yet it is **independent of the parameterization scheme $f$**.
>     Therefore our Theorem 3.4 (in Page 6) is also effective for more complex structures.
>     The reparameterization only change the recurrent weights' gradients, the gradients of other weights will not be influenced.
>
>     We include further numerical experiments (see Appendix D, page 22-24) on the image classifications and associative recalls to demonstrate the effectiveness of stable reparameterizations.
>     It can be seen the stable reparameterizations guarantee a good performance while the case without reparameterization fail to achieve convergence.
>     As we maintain the same initializations of the weights, the only difference across these models are the parameterization schemes of the recurrent weights $\Lambda$.
>
>
> 2. `It is not clear that why the "best" reparameterization should satisfy that the gradient is Lipshitz to the weight.`
>
>     We propose the "best" reparameterization to further improve the optimization stability.
>     The **intuitive** explanation is that a function with "gradient is Lipschitz to the weight" is more similar to the quadratic functions.
>     Therefore this property is important for the optimization.
>
>     A theoretical explanation will be given as follow:
>     For non-convex optimizations, the Lipschitz-smooth property is a commonly used assumption to achieve the convergence of the models [1].
>     Without Lipschitz-smooth property, function $f(x) = \sqrt{x}$ cannot converge to the optimal solution $x^*=0$ using fixed positive learning rate $\eta$.
>
>     The criterion "gradient is Lipschitz to the weight" is a **necessary condition** for the stability in the following sense (We include the detailed proof in Appendix F, page 25):
> - Consider functions $f(x) = x^4$, the gradient function $g(x)=4x^3$ does not have a global Lipschitz coefficient for all input values $x$. Therefore for any positive fixed learning rate $\eta$, there exists an initial point $x_0$ such that the convergence cannot be achieved via the gradient descent step.
>     $$x_{k+1} = x_k - \eta g(x).$$
>     Usually a smaller learning rate and a fine-tuned learning rate scheduler is necessary for the  convergence at that $x_0$.
> - Consider functions $f(x) = x^2$, the gradient function $g(x) = 2x$ is associated with a Lipschitz constant $L=2$. Then the same gradient descent step converges for any $\eta \leq \frac{1}{L} = \frac{1}{2}$ and any initial point $x_0$.
> - As can be seen in the above two examples, the criterion **"gradient is Lipschitz to the weight" is associated with the convergence under large learning rate**.
>     As the use of larger learning rate is usually associated with faster convergence [2], smaller generalization errors [3], we believe the Lipschitz criterion is a suitable stability criterion for the measure of optimization stability.
> - The gradient-to-weight ratio evaluated in Figure 4(a) is a numerical verification of our Theorem 3.4.
>     The gradients scale of stable reparameterizations are milder therefore less susceptible to the well-known issue of exploding or vanishing gradients [4,5].
>
>
>
> Questions:
> 1. `As the numerical examples mainly show the results on gradient to weight ratio, how can you justify that this quantity is related with better training?`
>
>     Kindly refer to the response provided for point 2 of the weaknesses, particularly the last paragraph, for further clarification on this question.
>
> 2. `Can this analysis be extended to more complex models such as multi-layer RNN or transformer-based RNN?`
>
>     As we mentioned in reply to point 1 of the weaknesses, our analysis can be extended in the following sense:
>     When we evaluate the gradients of the recurrent weights in complex multi-layer models, the state-space model layers also satisfy the same gradient bounds from the Theorem 3.4.

---

> ### Author Response · Authors · 2023-11-22
>
> References
>
> [1] Allen-Zhu, Zeyuan, Yuanzhi Li, and Zhao Song. "A convergence theory for deep learning via over-parameterization." In International conference on machine learning, pp. 242-252. PMLR, 2019.
>
> [2] Smith, Leslie N., and Nicholay Topin. "Super-convergence: Very fast training of neural networks using large learning rates." In Artificial intelligence and machine learning for multi-domain operations applications, vol. 11006, pp. 369-386. SPIE, 2019.
>
> [3] Li, Yuanzhi, Colin Wei, and Tengyu Ma. "Towards explaining the regularization effect of initial large learning rate in training neural networks." Advances in Neural Information Processing Systems 32 (2019).
>
> [4] Bengio, Yoshua, Patrice Simard, and Paolo Frasconi. "Learning long-term dependencies with gradient descent is difficult." IEEE transactions on neural networks 5, no. 2 (1994): 157-166.
>
> [5] Hochreiter, Sepp. "The vanishing gradient problem during learning recurrent neural nets and problem solutions." International Journal of Uncertainty, Fuzziness and Knowledge-Based Systems 6, no. 02 (1998): 107-116.

---

> ### Comment · Reviewer_4Tez · 2023-11-22
>
> It seems that the crucial theorem is that the memeory function $\rho(t)$ can be approximated with recurrent weights $\Lambda$ when the model is linear RNN. Wouldn't such $\Lambda$ be unobtainable when the model is not simple linear RNN?

---

> ### Author Response · Authors · 2023-11-23
>
> Thank you for your prompt response.
>
> Theorem 3.3 is established for the approximation of linear functional with linear RNNs.
> When the target is nonlinear functional and models are multi-layer state-space models with layer-wise nonlinearities, the same result holds in the following sense:
> Let $\mathbf{H}$ be the target sequential relationship, $\widehat{\mathbf{H}}$ be the theoretical optimal model and $\widetilde{\mathbf{H}}$ be the perturbed model, then by the triangular inequality the stable approximation of the target $\mathbf{H}$ can be achieved if the optimal model $\widehat{\mathbf{H}}$ can be stably approximated.
> $$|| \mathbf{H} - \widetilde{\mathbf{H}}|| \leq || \mathbf{H} - \widehat{\mathbf{H}}|| + || \widehat{\mathbf{H}} - \widetilde{\mathbf{H}}||.$$
> As the optimal model comes with a specific representation in $\Lambda^*$, therefore apply the result from linear RNNs can give us the stable approximation of optimal model $\widehat{\mathbf{H}}$ with perturbed models.

---

### Official Review · Reviewer_fvkX · 2023-11-02

**Soundness:** 3 good
**Presentation:** 3 good
**Contribution:** 3 good
**Rating:** 6
**Confidence:** 3

**Summary:**

The work proposed a class of reparameterization techniques that lifts the memory limitations in SSM. The authors provide both theoretical analysis and empirical evaluation on the proposed approach.

**Strengths:**

The paper is well organized with detailed theoretical analysis and empirical evaluation. The overall workflow is pretty easy to follow.

The authors demonstrate that the model structure of state-space models does not address the curse of memory phenomenon, and proposed the stable reparameterization to tackle the issue, While the reviewer didn't checked every detail, the derivation looks to be concrete.

**Weaknesses:**

The empirical evaluation on the synthetic dataset and language model seems are mostly on training behavior, e.g. decrease on training loss, improvement on stability of the training curve etc. Could the proposed approach concretely improve the testing performance? More evaluation on model performance on testing data is needed.

The author mentioned in multiple places that the theorems are established for the shallow case. Better to make a clarification on how shallow it is and why the same don't establish for deeper cases.

In introduction, further illustration and demonstration on the disadvantage of exponential decay in memory is needed, as it is the main problem tackled in the work.

**Questions:**

Could the proposed approach concretely improve the testing performance?

---

> ### Author Response · Authors · 2023-11-22
>
> We thank the reviewer for the positive feedback.
>
> Weaknesses:
> 1. `The empirical evaluation on the synthetic dataset and language model seems are mostly on training behavior, e.g. decrease on training loss, improvement on stability of the training curve etc. Could the proposed approach concretely improve the testing performance? More evaluation on model performance on testing data is needed.`
>
>     Thank you for pointing out the missing validation/test issue.
>     In this paper, we mainly study the effects of parameterization on the approximation and optimization.
>     We show in Appendix D that with stable reparameterizations, the reparameterized models' performances over language model, associative recall and image classification are in general better than the models' without reparameterization.
>     Despite the "best" reparameterization does not always present the optimal numerical performance, we further construct an example to show it is optimal in stability (Appendix D.2, page 24).
>     The additional numerical results are summarized in the tables in reply-to-all.
>
>
> 2. `The author mentioned in multiple places that the theorems are established for the shallow case. Better to make a clarification on how shallow it is and why the same don't establish for deeper cases.`
>
>     The Theorem 3.4 is given in the shallow case but the gradient norm bounds hold for the deeper models.
>
>     As the reparameterization of state-space models also change the gradient of state-space models, so the gradient analysis holds for the state-space model layer and the gradient of recurrent weights in more complex structures will also be bounded by a parameterization-dependent term in Equation (12), page 6.
>     Therefore the stable reparameterization will also be effective to remove the curse of memory in more complex structures.
>     We include the discussion in Q1 of reply-to-all.
>
>
> 3. `In introduction, further illustration and demonstration on the disadvantage of exponential decay in memory is needed, as it is the main problem tackled in the work.`
>
>     Thank you for pointing out this writing issue.
>
>     The exponential decay in memory means the models' outputs have a short context in the sense that changing the inputs $x_{(-\infty,T-l]}$ before context length $l$ has little influence on the outputs $y_T$. For some constant $a$ and $b$,
>     $$|y_T - y_T'| \leq a * e^{-\beta l} |x_{T-l} - x_{T-l}'|.$$
>     Therefore the models cannot capture long-term relationships [1].
>     We have revised the introduction to emphasize the importance of learning long-term memory.
>
>
>
> Questions:
> 1. `Could the proposed approach concretely improve the testing performance?`
>
>     We have answered this question in the reply to point 1 of the weaknesses.
>     We show the stable reparameterizations improve the learning of long-term memory targets in both approximation sense (Figure 1, Page 5) and optimization sense (Appendix D, Page 22-24).
>
>
>
> ---
> References
>
> [1] Zhong Li, Jiequn Han, E. Weinan, and Qianxiao Li. "On the Curse of Memory in Recurrent Neural Networks: Approximation and Optimization Analysis." In International Conference on Learning Representations. 2020.

---

### Official Review · Reviewer_uWCH · 2023-11-03

**Soundness:** 1 poor
**Presentation:** 1 poor
**Contribution:** 2 fair
**Rating:** 5
**Confidence:** 3

**Summary:**

This paper analyzes how SSM-based models approximate target sequences. It proposes a simple criterion based on gradient norm scales to improve the implicit parametrization for the eigenvalue of real-valued SSM, first on a synthetic approximation task, then on a gated-convolution model.

**Strengths:**

* Choosing an appropriate implicit parametrization for SSMs is quite important in practice, and this paper provides a criterion to rank them.
* I found some of the theoretical connections quite interesting e.g., how the authors use Volterra series to express these models.

**Weaknesses:**

* My main concern with this paper is the disconnect between theory and experimental results: the authors train few small-scale Hyena-SSM model on wikitext, and then attempt to explain how the ranking in training loss corresponds to a ranking with the proposed gradient-norm scale. There are no attempts to perform multiple runs, try in different applications, or verify whether this hypothesis holds with different hyperparameters. In fact, Appendix D shows the rankings change completely by tweaking the learning rate.

**Questions:**

* The experiments rely on exploring different options for the implicit parametrization $f(w)$ of the eigenvalues of a real-valued SSM. Have you considered using small hypernetworks, or alternative parametrizations? Why parametrize only the poles of the SSM implicitly, and not also the residues?
* Have you considered the effect of the entire architecture block, composed of gating and the SSM, on the choice of parametrization? Could that inform a better metric that better correlated with performance in practice?
* Can you provide some downstream evaluation of the language model, or at the very least validation loss?


Some nitpicks:
* Eq (2) does not correspond to practical instantiation of SSM-based models, which have linear readout ($c^T h_t$), then a pointwise shallow MLP (either with gating or without).

*UPDATE*: The authors have provided some clarifications and additional numerical experiments. While I do not think the experiments are conclusive, the paper puts forward a compelling theoretical argument that could produce improvements to the parametrization of SSM layers. I have raised my score.

---

> ### Author Response · Authors · 2023-11-22
>
> We thank the reviewer for their thoughtful comments and constructive feedback.
>
> Summary:
>
> The experiment is not established via gated-convolution model.
>     We use a hyena-based state-space model which parameterizes the recurrent weights using S5 layer instead of the Hyena's implicit convolution form.
>
>
>
> Weaknesses:
> 1. `My main concern with this paper is the disconnect between theory and experimental results: the authors train few small-scale Hyena-SSM model on wikitext, and then attempt to explain how the ranking in training loss corresponds to a ranking with the proposed gradient-norm scale. `
>
> Here are the connections between theory and experimental results:
>
> 1. We study the stability of state-space models from the parameterization viewpoint.
> 2. For **approximation** theory, we point out the importance of stable reparameterization in learning long-term memory.
>     We prove that stable reparameterization is **necessary** (Theorem 3.1, Page 5) and **sufficient** (Theorem 3.3, Page 6) for the long-term memory learning.
>     The necessity is proven in the sense that without reparameterization, the state-space models can only stably approximate targets with exponential memory decay.
>     The sufficiency is proven in the sense that with stable reprameterizations, the linear RNN can stably approximate the linear functionals with **any** memory decay.
>     **We identify a family of stable reparameterizations** (Definition 3.2) that can achieve the stable approximation of long-term memory targets:
>     For some continuous function $g: [0, \infty) \to [0, \infty), g(0)=0$,
>     $$\sup_w \left[ |f(w)| \sup_{|\tilde{w} - w| \leq \beta} \int_0^\infty \left | e^{f(\tilde{w}) t}  - e^{f(w) t} \right | dt \right ] \leq g(\beta).$$
>     This family includes a large set of stable reparameterizations.
>     In Appendix B.4 Remark B.3 (Page 17), we show the commonly used parameterizations such as the exponential and softplus parameterization are all stable parameterizations.
> 3. Building upon the above theoretical results, it's important to note that there exist alternative reparameterization methods, which, though less explored, are equally capable of achieving a stable approximation of long-term memory.
>     This leads us to a pertinent inquiry: What constitutes the **optimization criterion** for selecting the most stable reparameterization approach in practical applications?
>     The Lipschitz smoothness of the gradient is commonly adopted as an assumption for non-convex optimization, therefore we propose the Lipschitz continuity of gradient in Equation 11 (Page 6) as an optimization criterion for good stability.
>     Further details regarding the motivation for this optimization criterion are comprehensively outlined in Appendix F, on page 23.
>     The gradient norm bounds and the Lipschitz continuity criterion allow us to compare different stable reparameterizations and find the **"best" in the criterion of Lipschitz continuity**.
>
>     (**Wikitext as verification**)
>     The training of Hyena-S5 model with 120M parameters on Wikitext is used as a verification for the approximation and optimization properties in Theorem 3.4.
>     We show the initial gradient-over-weight scale in Figure 4(a) and corresponding training loss in Figure 4(b).
>
>     (**Numerical verification of "best" parameterization**)
>     The experiments on Wikitext only present the performance of stable reparameterizations over unstable reparameterizations.
>     We further verify the optimization stability of "best" reparameterization in the following extreme setting.
>     We construct a large scale language model with 3B parameters and train with larger learning rate (lr=0.01).
>     As can be seen in the following table, the only convergent model is the model with "best" reparameterization.
>     We emphasize that the **only difference between these models are the parameterization schemes** for recurrent weights.
>     Therefore the best reparameterization is the most **stable** parameterization.
>     (We repeat the experiments with different seeds for three times. The detail of the experiment is given in Appendix D.2, page 24)
>
> | Reparameterizations | Convergent / total experiments |
> |------------|-----|
> | **"Best"** | 3/3 |
> | Exp        | 0/3 |
> | Softplus   | 0/3 |
> | Direct     | 0/3 |

---

> ### Author Response · Authors · 2023-11-22
>
> `There are no attempts to perform multiple runs, try in different applications, or verify whether this hypothesis holds with different hyperparameters. In fact, Appendix D shows the rankings change completely by tweaking the learning rate.`
>
> This is a paper on the study of theoretical properties of parameterizations.
> The results on synthetic task and language models are numerical verification of the Theorems 3.1, 3.3 and 3.4.
>
> We further provide additional numerical experiments including associative recall (over long sequences with length 130K) and image classification (MNIST / CIFAR10) in reply-to-all.
> It can be seen the stable reparameterizations are all better than the case without stable reprameterization.
> Also, the best reparameterization is better than the exponential and softplus parameterization in the MNIST and CIFAR10 tasks.
>
>
>
> Questions:
>
> 1. `The experiments rely on exploring different options for the implicit parametrization  of the eigenvalues of a real-valued SSM. Have you considered using small hypernetworks, or alternative parametrizations? Why parametrize only the poles of the SSM implicitly, and not also the residues?`
>
>     Thank you for the interesting proposal.
>     The approximation and optimization results of our work have close connections with the suggestion in the following sense:
>     - First, the result in Equation 9 gives a sufficient condition for stable approximation.
>         This condition identifies a family of "feasible hypernetworks" for learning long-term memory.
>         Therefore the parameterization with the constraint from Equation (9) gives a feasible set for further optimization.
>     - Second, the Lipschitz criterion (Equation 11) is taken as the specific target for optimization stability in this paper.
>         Our "best" parameterization is proposed based on the optimization stability criterion - "Lipschitz continuity criterion".
>         We solve the ordinary differential equation and achieves a low-freedom parameterization: $f(w) = \frac{1}{aw^2+b}$ with only two degress of freedom.
>         This is consistent with the optimization over hypernetwork idea that your suggested.
>     - Future promising direction includes using hypernetworks and finding suitable reparameterization with gradient-based optimization.
>         With constraints from Equation 9 and new objectives, we can still optimize the hypernetwork, which is out of scope of current work.
>
>     The reason we only parameterize the poles of the SSM implicitly is based the empirical evidence that the output of the model for large time is mostly sensitive to the value of poles [1,2,3,4]. and the gradient of the state-space model parameters
>     The state-space models accumulated the hidden state with linear system $h_{k+1} = \Lambda h_k + U x_k$.
>     **When the sequence length $T$ is large, $h_T = \sum_{i=0}^{T-1} \Lambda^{T-k} U x_k$, even a minor difference of $\Lambda$ will result in a significant change of hidden state $h_T$.**
>
>     The selection of parameters for other layers is equally important, though they are less prone to stability issues.
>
>
> 2. `Have you considered the effect of the entire architecture block, composed of gating and the SSM, on the choice of parametrization? Could that inform a better metric that better correlated with performance in practice?`
>
>     The entire architecture block is usually complex with different stack of normalization, residual connection and dropouts.
>
>     Our analysis can be generalized to the multi-layer state-space models of complex architectures:
>     Consider the gradient of the recurrent weights in a complex architecture, we still have the same gradient norm bound as characterized by the Theorem 3.4.
>     In particular, the gradient norm bound only depends on the target function, current model function, and the parameterization-dependent scale $\frac{|f'(w)|}{f(w)^2}$.
>     The detailed discussion is presented in the Q1 of reply-to-all.

---

> ### Author Response · Authors · 2023-11-22
>
> 3. `Can you provide some downstream evaluation of the language model, or at the very least validation loss?`
>
>     This paper primarily presents a theoretical exploration of parameterization properties.
>     We provide the validation loss and perplexity with the standard error for the wikitext103 language modelling.
>
>     | Reparameterizations | Train ppl | Train loss | Test ppl | Test loss |
>     |-------------|-----------|------------|----------|-----------|
>     | Best        | 17.182    | 2.844      | 20.811   | 3.035     |
>     | Exp(S5)     | 15.721    | 2.755      | 20.218   | 3.007     |
>     | Softplus    |**14.570** | **2.679**  |**20.136**| **3.003** |
>     | Direct      | 18.916    | 2.940      | 28.167   | 3.338     |
>
>     Due to the small size of wikitext dataset and limitation of computing resources, we do not carry out the downstream evaluation of the language models.
>     This evaluation will be part of future numerical evaluation directions.
>
>
>
> 4. `Eq (2) does not correspond to practical instantiation of SSM-based models, which have linear readout ($c^T h_t$), then a pointwise shallow MLP (either with gating or without).`
>
>     The Equation 2 corresponds to the practical instantiation of SSM-based models in the following sense:
>     Any practical instantiation of SSM-based models can be implemented as a stack of Eq (2).
>     The pointwise shallow MLP can be realized with state-space model layer with layer-wise nonlinearity by setting recurrent weights $W$ to be 0.
>     $$h_{k,l+2} = U_1 \sigma (U_2 h_{k, l} + b_2) + b_1.$$
>     We give a graphical demonstration in Appendix E, page 24.
>     Therefore, the Equation 2 is given in a simplified form without sacrificing the expressiveness of the multi-layer state-space models.
>     The results established from Theorem 4.1, 4.3 and 4.4 also hold for the practical versions.
>
>
>
> ---
>
> References
>
> [1] https://github.com/HazyResearch/state-spaces/blob/main/example.py. The learning rate of recurrent weights is 0.001 while the learning rate for the rest is 0.01.
>
> [2] https://github.com/HazyResearch/safari/issues/28. Hyena architecture is said to be quite sensitive to a few hyperparameters.
>
> [3] https://wandb.ai/jimmysmith1919/S5_ICL/reports/Hyena-red-and-Hyena-S5-blue-on-WikiText-103--Vmlldzo0MTkwODEx?accessToken=pk0zw5w75uo1s4zkn3kh7koum902t4q2yzbm28xk0olzzgxuskoq0g1iyauixlob. The training of Hyena is highly sensitive to the learning rate. The S5 using the exponential parameterization has a relatively better stability.
>
> [4] https://arxiv.org/abs/2208.04933 A smaller learning rate (the SSM learning rate) is applied to the recurrent weights.

---

### Author Response · Authors · 2023-11-22
**Reply to all**

We thank all the reviewers for their constructive reviews.

## Common questions

1. Q1: Does our result (Theorem 4.4) hold for multi-layer complex models?

    A: Yes.
    The reparameterization technique can also be applied to sequential models with more complex structures in the following sense:
    If we consider the gradient of recurrent weights $\Lambda=f(w)$, the same gradient norm bounds hold for the complex structures.
        $$G_{f}(w) := \left | \frac{\partial \textrm{Loss}}{\partial w} \right | \leq C_{\mathbf{H}, \widehat{\mathbf{H}}_m} \frac{|f'(w)|}{f(w)^2}.$$
    The coefficient $C$ is a target-dependent and model-dependent coefficient, yet it is **independent of the parameterization scheme $f$**.
    Therefore our Theorem 3.4 (in Page 6) is also effective for more complex structures.
    As the reparameterization only change the recurrent weights' gradients, the gradients of other weights will not be influenced.

2. Q2: Does our theory indicate improved test performance?

    The focus of this paper is on the effect of parameterization on the approximation and optimization.
    In particular, we study the stability as the models face difficulty in stability in both the approximation and optimization.
    In the following attached tables and Appendix D of the revised paper, we show the improved test performance of stable parameterization over the models without stable reparameterizations.



## Additional numerical results
*Associative recall* (Datasets with different sequence lengths and vocabulary sizes)

|Reparameterizations| Train acc (Vocab size=10)|Test acc (Vocab size=10)|Train acc (30)|Test acc (30)|
|-------------|------------------|-----------------|----------------|---------------|
| **Best**    | **57.95**            | **99.8**        | **53.57**          | **100**       |
| Exp(S5)     | 54.55            | **99.8**        | **53.57**          | **100**       |
| Clip        | 50.0             | 76.6            | 13.91          | 9.4           |
| Direct      | 43.18            | 67.0            | 16.59          | 5.6           |

*MNIST* (Repeated three times with standard deviation.)

|Reparameterizations| Train acc | Train loss | Test acc  | Test loss |
|-------------|---------------------|--------------------|-----------------|------------------|
| **Best**    | **99.31(0.0264)**   | 0.02269(0.00121)   | **99.3**(0.153) | 0.02501(0.00519) |
| Exp(S5)     | 99.26(0.241)        | 0.02305(0.000694)  | 99.21 (0.0321)  | 0.02546(0.00118) |
| Softplus    | 99.14(0.036)        | 0.02337(0.00132)   | 99.19 (0.0458)  | 0.02682(0.000769)|
| Direct      | 9.84(0.0753)        | Diverged           | 10.09 (0.742)   | Diverged         |

*CIFAR10* (Repeated three times using different seeds with standard deviation in parentheses.)

|Reparameterizations| Train acc | Train loss | Test acc | Test loss |
|-------------|-----------------|--------------------|-----------------------|--------------------|
| **Best**    |**61.97(0.500)** | **1.059(0.0155)**  | **66.41(0.156)**      | **0.9430(0.00248)**|
| Exp(S5)     | 61.77 (0.690)   | 1.062(0.0141)      | 65.87(0.136)          | 0.9487(0.00983)    |
| Softplus    | 61.53 (0.716)   | 1.0683(0.0142)     | 65.84(0.05)           | 0.9575(0.00189)    |
| Direct      | 9.988 (0.032)   | Diverged           | 9.736(8.533E-05)      | Diverged           |



## Summary of the revisions
Below we present a summary of the revisions.
In the paper, the revision has been marked in blue.

1. In Appendix D (Page 22-24), we add the additional numerical results for 'stability verification of "best" reparameterizations', comparison of stable reparameterizations over associative scans and image classifications.
2. In Appendix E (Figure 7), we give a graphical demonstration of state-space models as stack of Equation (2).
    Therefore our theorems proved for simple formulation also extend to the complex architectures.
3. In Appendix F (Page 25), we give a detailed explanation of the motivation for gradient-to-weight Lipschitz criterion.
4. In Appendix G (Page 26), we include the "best" parameterization solution derived from the Lipschitz criterion ODE to ensure completeness.

---

### Meta-Review · Area_Chair_9GFc · 2023-12-09

**Metareview:**

The paper explores the long-term memory learning capabilities of SSMs. The authors claim that standard SSMs, without reparameterization, exhibit memory limitations similar to traditional RNNs, primarily in stably approximating relationships with exponential decaying memory. They introduce a reparameterization technique for SSMs, aiming to extend their memory capabilities. The authors argue that this not only improves the models' approximation abilities but also enhances optimization stability.

Strengths: The paper presents a significant theoretical analysis of the memory limitations in state-space models and proposes a novel solution through reparameterization techniques.

Weaknesses:
It was pointed out by reviewers that there exists a gap between the theoretical claims and the experimental results which I agree. Some experimental setups (like the Hyena-SSM model on Wikitext) were not sufficiently aligned with the theoretical framework. Moreover, The empirical evaluation mainly focuses on training behavior and lacks a comprehensive assessment of testing performance across various domains. A more elaborate empirical evaluation of the method is needed before the paper becomes ready for acceptance.

**Justification For Why Not Higher Score:**

The paper presents a considerably good theoretical contribution with potential practical implications in sequence modeling. Despite the authors' efforts to address the concerns, there remain essential gaps in empirical validation, particularly in demonstrating the applicability of their approach to complex models and in diverse testing scenarios. I believe the paper will benefit from another round of review to show the practical benefits of the theoretical results.

**Justification For Why Not Lower Score:**

N/A

---

### Decision · Program_Chairs · 2024-01-16

Reject